



# Wildfire influences on the variability and trend of summer surface ozone in the mountainous western United States

Xiao Lu[1], Lin Zhang[1], Xu Yue[2], Jiachen Zhang[3], Daniel A. Jaffe[4], Andreas Stohl[5], Yuanhong Zhao[1],

5    Jingyuan Shao[1]

[1]Laboratory for Climate and Ocean-Atmosphere Sciences, Department of Atmospheric and Oceanic

Sciences, School of Physics, Peking University, Beijing 100871, China

[2]School of Forestry and Environmental Studies, Yale University, New Haven, Connecticut 06511, USA

10   [3]Department of Civil and Environmental Engineering, Viterbi School of Engineering, University of

Southern California, Los Angeles, CA 90089, USA

[4]School of Science, Technology, Engineering and Math, University of Washington Bothell WA and

Department of Atmospheric Sciences, Seattle WA 98011, USA

[5]Norwegian Institute for Air Research, 2007 Kjeller, Norway

*Correspondence to:* Lin Zhang (zhanglg@pku.edu.cn)



**Abstract.** Increasing wildfire activities in the mountainous western US may present a challenge for the

region to attain a recently revised ozone air quality standard in summer. Using current Eulerian

chemical transport models to examine the wildfire ozone influences is difficult due to uncertainties in

fire emissions, inadequate model chemistry and resolution. Here we quantify the wildfire influence on

the ozone variability, trends, and number of high MDA8 (daily maximum 8-h average) ozone days over

this region in summers (June, July and August) 1989-2010 using a new approach. We define a Fire

Index using retroplumes (plumes of back-trajectory particles) computed by a Lagrangian dispersion

model (FLEXPART), and develop statistical models based on the Fire Index and meteorological

parameters to interpret MDA8 ozone concentrations measured at 13 Intermountain West surface sites.

We show that the statistical models are able to capture the ozone enhancements by wildfires and give

results with some features different from the GEOS-Chem Eulerian chemical transport model. Wildfires

enhance the Intermountain West regional summer mean MDA8 ozone by 0.3-1.5 ppbv (daily episodic

enhancements reach 10-20 ppbv at individual sites) with large interannual variability, which are

strongly correlated with the total MDA8 ozone. Wildfires also contribute 15% of the measured

increasing but statistically insignificant trends of 0.14-0.19 ppbv year$^{-1}$ in 1989-2010. We find large fire

impacts on the number of exceedance days; for the 13 CASTNet sites, 31% of the summer days with

MDA8 ozone exceeding 70 ppbv would not occur in the absence of wildfires.



## 1 Introduction

Ozone is a secondary air pollutant that exerts negative effects on human health and vegetation, and is also a short-lived greenhouse gas with a positive radiative forcing of 0.40 (0.20 to 0.60) W m$^{-2}$ (Shindell et al., 2013; Stevenson et al., 2013; Stocker et al., 2013; Cooper et al., 2014; Monks et al., 2015). Tropospheric ozone is generated through sunlight driven chemical oxidation of CO, CH$_4$, and other non-methane volatile organic compounds (NMVOCs) in the presence of nitrogen oxides (NO$_x$=NO+NO$_2$). It can also be transported from the stratosphere. In October 2015, the US Environmental Protection Agency (EPA) lowered the National Ambient Air Quality Standard (NAAQS) for ozone, defined as the annual fourth-highest daily maximum 8-h average (MDA8) concentration averaged over three years, from 75 ppbv to 70 ppbv (US EPA, 2015). Attaining this lower ozone air quality standard places new challenges for the US states (Cooper et al., 2015).

Ozone over the mountainous western US (US Intermountain West), extending between the Sierra Nevada/Cascades to the west and the Rocky Mountains in the east, has recently drawn an increasing attention (Cooper et al., 2015; Lin et al., 2015a). Unlike in the eastern US, where NO$_x$ emission controls have led to ozone declines, surface ozone concentrations in the Intermountain West have been increasing in the 1990-2010 period most likely caused by rising background ozone (Jaffe et al., 2007; Cooper et al., 2012; Lin et al., 2015), although recent research suggests that these trends flatten out or



even reverse in the later decade (2000-2010) (Cooper et al., 2014; Simon et al., 2015; Strode et al.,

2015). The North American background ozone, defined by the US EPA as the surface ozone

concentration that would be present over the US in the absence of anthropogenic emissions from North

America (US EPA, 2006), is particularly high in the Intermountain West due to high elevation, arid

landscape, and frequent large-scale air subsidence (Fiore et al., 2002; McDonald-Buller et al., 2011;

Zhang et al., 2011; Emery et al., 2012; Dolwick et al, 2015). The background ozone includes ozone

contributed by anthropogenic emissions outside North America, e.g., over Asia and Europe (Zhang et al.,

2009; Cooper et al., 2010; Lin et al., 2012a), as well as natural sources such as lightning (Mueller et al.,

2011; Zhang et al., 2014), wildfires (Jaffe et al., 2008, 2013; Mueller et al., 2011; Zhang et al., 2014),

and stratospheric influxes (Lin et al., 2012b, 2015b; Zhang et al., 2014). A number of studies have

shown that model simulations considering rising Asian emissions and global methane can only explain

part of the observed increasing ozone trends in the western US (Fiore et al., 2009; Koumoutsaris et al.,

2012; Parrish et al., 2014).

Wildfires are potentially important sources of background ozone, as they emit large amounts of $NO_x$,

CO, and NMVOCs particularly in summer under hot and dry weather conditions conducive to ozone

formation. There is evidence that the frequency and intensity of wildfires in the western US have been

increasing from 1970s to 2005 driven by increasing temperatures and earlier snowmelt (Westerling et al.,



2006). The number of high-ozone days is shown to have a strong interannual correlation with wildfire

burned area over this region (Jaffe et al., 2008, 2012). However, quantifying ozone production in

wildfire plumes is complicated by various uncertainties including those in wildfire emissions, chemical

reactions, and variations in meteorology such as changes in temperature (Jaffe et al., 2012). Fire

emissions of ozone precursors vary significantly among different ecosystem types, biomass nitrogen

loads, and combustion efficiency (Andreae et al., 2001; Akagi et al., 2011). Ozone chemistry in fire

plumes shows strong non-linearity with observations of ozone over CO enhancements ($\Delta O_3/\Delta CO$) in

fire plumes ranging from -0.1 to 0.9 ppbv ppbv$^{-1}$ depending on plume ages, aerosol effects, and mixing

with urban emissions (Real et al., 2007; Jaffe et al., 2012; Singh et al., 2012; Parrington et al., 2013;

Baylon et al., 2014). Previous studies also suggested that rapid conversion of $NO_x$ to peroxyacetyl

nitrate (PAN) would limit ozone production near the fires (especially at low temperatures), but

decomposition of PAN could lead to additional ozone production further downwind of the fires

(Alvarado et al., 2010; Jaffe et al., 2012, 2013).

A standard approach to quantify the influence of a particular source on ozone concentrations is provided

by chemical transport models (CTMs) using the differences between model simulations with and

without this source. This Eulerian approach has been applied in numerous studies to examine ozone

from different sources based on global and regional CTMs (Pfister et al., 2007; Alvarado et al., 2010;



Grell et al., 2011; Jiang et al., 2012; Zhang et al., 2014). However, application of this approach to assess

wildfire ozone influences in the US Intermountain West is particularly challenging due to uncertainties

in wildfire emissions and model chemistry as well as limited model resolution (Zhang et al., 2014). Our

current understanding of wildfire influences on the variability and long-term trends of surface ozone is

rather limited (Jaffe et al., 2012; Fiore et al., 2014).

In this study, we propose a new approach to estimate the influence of wildfires on surface ozone

concentrations in the US Intermountain West. We define a Fire Index using the retroplumes (plumes of

back-trajectory particles) calculated by a Lagrangian particle dispersion model (FLEXPART) combined

with a daily high-resolution wildfire area burned dataset. We then develop multiple linear regression

(MLR) models to estimate surface ozone concentration as a function of the Fire Index and other

meteorological parameters, which allow us to separate the influences of wildfires and meteorology. We

apply this approach to interpret surface ozone concentrations measured at CASTNet (the Clean Air

Status and Trends Network) sites in the US Intermountain West during the summers (June, July and

August) 1989-2010, and to quantify wildfire influences on the ozone interannual variability, trends, and

exceedance days (MDA8 ozone>70 ppbv) over this region. The Lagrangian-based wildfire ozone

influences are also compared with those estimated by a Eulerian model (GEOS-Chem) to evaluate the

consistency and difference between the two.



## 2 Materials and Methods

### 2.1 Data description

We use measurements of ozone and organic carbon (OC) aerosols, meteorological parameters, as well as wildfire area burned data at daily temporal resolution. Hourly measurements of ozone as well as meteorological parameters including surface temperature, wind speed, relative humidity (RH), and solar radiation are accessed from CASTNet (http://www.epa.gov/castnet). We focus on measurements at 13 CASTNet sites in the US Intermountain West for 1989-2010 (Figure 1 and Table 1). Most CASTNet sites have ozone measurements for the 22-year period except for Mesa Verde National Park (NP) (MEV), Great Basin NP (GRB), Canyonlands NP (CAN), and Big Bend NP (BBE) (since 1995), and Petrified Forest (PET) (since 2003). The Yellowstone NP (YEL) site experienced monitor relocation in 1996, and we access the 1989-1995 measurements at the earlier YEL site from the National Park Service (NPS) following Jaffe et al. (2007) and Cooper et al. (2012).

In addition, we use hourly ozone measurements from 1990-2010 at the Salt Lake City (SLC) urban site (data available at https://www3.epa.gov/airdata/) for comparison with the CASTNet background sites and the previous work of Jaffe et al. (2013). Measurements of OC aerosol are from collocated sites of the Interagency Monitoring of Protected Visual Environments (IMPROVE,



http://vista.cira.colostate.edu/improve/). OC aerosol concentrations are 24-hour averages measured

every 3 days.

We also use the daily wildfire area burned data over North America for 1989-2010 developed by Yue et

al. (2013) that has a 0.5 °×0.5 °horizontal resolution. This inventory is constructed using the

inter-agency fire reports from the national Fire and Aviation Management Web application system

(FAMWEB, https://fam.nwcg.gov/fam-web/), and applied with a daily scaling factor for the duration of

each fire event based on local meteorological variables (Yue et al., 2013).   The total areas burned in the

Intermountain West range from 90,000 to 2,000,000 hectares (ha) in the summers 1989-2010 with a

large spatial and interannual variability. This wildfire area burned inventory has been used in Zhang et

al. (2014) and was able to capture the episodic enhancements of OC aerosol concentrations measured in

the Intermountain West for the summers 2006-2008.

**2.2  Fire Index calculation with the FLEXPART model**

Jaffe et al. (2008) previously identified the impacts of wildfires on ozone at a measurement site using

values of monthly wildfire area burned or carbon burned within a certain region around the site (e.g.,

10 °×10 °or 5 °×5 °). This fire indicator generally ignores the variable influence of transport of fire

plumes to the site. For instance, a fire downwind of the measurement site, even one burning in the





immediate vicinity, would not influence the site. Here we propose a new fire indicator using 5-day

retroplumes simulated by the FLEXPART Lagrangian particle dispersion model and the daily wildfire

area burned inventory mentioned above. A retroplume consists of a large number of back trajectory

particles that are released from a particular receptor location (Cooper et al., 2005). We use FLEXPART

version 8.02, which is first described by Stohl et al. (2005) and has been applied to examine transport of

ozone (Cooper et al., 2010) and radionuclides across the Pacific Ocean (Stohl et al., 2012). FLEXPART

simulates the long-range and mesoscale transport, diffusion, dry and wet deposition of gases or particles

(Stohl et al., 2005). It is driven by the National Center for Environmental Prediction (NCEP) Climate

Forecast System Reanalysis (CFSR) data with 1-hour temporal resolution, 0.5 °×0.5 ° horizontal

resolution, and 37 vertical levels extending from the surface to 1 hPa.


For each day at a receptor site, FLEXPART was run in backward mode, with 250,000 particles released

at the site location at a constant rate during the 24 hours. The particles are set to have an e-folding time

of 5 days (mean lifetime of ozone in the Intermountain West as shown in Fiore et al. (2002)), and

trajectories of these particles are calculated backwards for 5 days (120 hours), together tracing the

retroplume of the air arriving at the site. The model outputs are in the same 0.5 °×0.5 ° horizontal

resolution as the wildfire area burned data, and are hourly residence times of the particles in each grid

cell. The residence time provides a quantitative measure of the sensitivity of the simulated mixing ratio



at the site location to emission input (Stohl et al., 2003; Cooper et al., 2010). In total, we have computed

over 28000 FLEXPART retroplumes for the 13 Intermountain West CASTNet sites and SLC site for the

summers 1989-2010.

We then define a Fire Index (FI) as the product of daily FLEXPART residence time integrated from the

surface to 5 km and daily wildfire area burned, in unit of s•ha. We use 5 km in the vertical because fire

emissions are often lifted to above the planetary boundary layer (Sofiev et al., 2013) and, as shown in

Table S2, it provides best correlations with the OC aerosol concentrations. The sum of Fire Index over

the 5-day period is defined as Total Fire Index (TFI). The formulas are given as:

$$\text{FI}(n) = \sum_i \sum_j E_{\text{fire}(i,j,n)} \times t_{r(i,j,n)} \tag{1}$$

$$\text{TFI} = \sum_{n=1}^{5} \text{FI}(n) \tag{2}$$

Here $E_{\text{fire}(i,j,n)}$ is the wildfire area burned in the model grid cell $i$ (longitude) and $j$ (latitude) on day $n$,

$t_{r(i,j)}$ is FLEXPART calculated daily residence time, and $n$ defines the backward day in the 5-day

period. Figure S1 shows an example of Fire Index for the site CAN on July 14, 2006. In this case, the

particles are released on July 14 (day $n$=1) in the FLEXPART model, and daily residence time is

calculated backwards for 5 days (July 10-14). FI(5) then represents the product of residence time on

July 10 and wildfire areas burned on that day. TFI as the sum of FI(1)-FI(5) estimates the total impact of

wildfires during the 5 days for that site and day.



## 2.3 Multiple linear regression model

We build multiple linear regression (MLR) models of summer ozone concentrations for the 13

CASTNet sites and SLC site using Fire Index and meteorological parameters as predictors. This method

has been previously used to identify the meteorological factors determining concentrations of

particulate matter or ozone (Camalier et al, 2007; Tai et al., 2010, 2012; Jaffe et al., 2013). Here we use

the metric of daily maximum 8-hour average (MDA8) ozone concentration, as it is the regulatory form

of the NAAQS. A total of 28 meteorological parameters are considered in the MLR models including

those measured at surface and from NCEP data. (Table 1 and Table S1) Some of these meteorological

variables, such as surface temperature, relative humidity, and upper level winds, have been shown

before to be correlated with surface ozone in the western US (Jacob et al., 2009; Rasmussen et al., 2012;

Jaffe et al., 2013).

Wildfire ozone enhancements are sensitive to plume ages. As summarized in Jaffe et al. (2012),

$\Delta O_3/\Delta CO$ values in wildfire plumes show distinct differences for plume ages of 1-2 days (average 0.018

ppbv/ppbv) versus 3-5 days (average 0.15 ppbv/ppbv). Thus instead of using TFI, we separate it to $FI_s$

($FI(1)+FI(2)$) and $FI_l$ ($FI(3)+FI(4)+FI(5)$) in the MLR models. We also include the square root of $FI_s$

and $FI_l$ ($SqrFI_s$ and $SqrFI_l$) as variables in the regression model to at least partly account for the





non-linearity of ozone chemistry in wildfire plumes. We do not use the natural logarithm form of FI in

MLR, because many of the FI values are zero that would cause invalid values in the regression.

The MLR models can be described as:

$$y = \alpha_1 \times \text{FI}_s + \alpha_2 \times \text{FI}_l + \beta_1 \times \text{SqrFI}_s + \beta_2 \times \text{SqrFI}_l + \sum_{p=1}^{m} \gamma_p \times \text{met}_p + c \ (3)$$

Here $y$ is MDA8 ozone concentration, $\alpha, \beta, \gamma$ are the regression coefficients, met denotes the $m$

meteorological parameters included, and $c$ is the constant term. We then estimate ozone enhancements

from wildfires and we refer it as MLR wildfire ozone, following:

$$y_{\text{fire}} = \alpha_1 \times \text{FI}_s + \alpha_2 \times \text{FI}_l + \beta_1 \times \text{SqrFI}_s + \beta_2 \times \text{SqrFI}_l \qquad (4)$$

The remaining components define the contribution from other variables such as meteorology and other

sources:

$$y_{\text{nofire}} = \sum_{p=1}^{m} \gamma_p \times \text{met}_p + c \qquad (5)$$

To further account for the nonlinear ozone response to wildfire emissions, we divide the ozone records

for each site into three subsets based on their TFI values: subset with TFI=0; subsets with the lower 50%

and upper 50% TFI values (with TFI=0 excluded). In this way we are able to quantify potentially

different ozone drivers under high vs. low wildfire conditions. The MLR models as described above are

applied to each subset.



Prior to performing the regression, we calculate correlations among ozone and all predictors and remove

those factors that show weak correlation with ozone but strong dependence on other predictors. To

minimize the collinearity in the MLR model, we also apply the stepwise regression method, i.e., for

each step the model selects the most powerful and significant ($p<0.05$) predictor explaining the residual,

and removes predictors with insignificant influence ($p > 0.1$) (Field et al., 2009). We acknowledge that

including FI and meteorological parameters in the MLR models inevitably leads to some degree of

collinearity. A measure of it is called tolerance (calculated as percent of variance in the predictor that

cannot be accounted for by the other predictors) or variance inflation factors (VIF, the inverse of

tolerance), with VIF values greater than 10 suggesting a strong collinearity (Field et al., 2009). Our

MLR models for all sites (Section 3) show tolerable VIF values (<5), supporting our approach described

above to limit the collinearity.

## 2.4 The GEOS-Chem model simulations

We further conduct GEOS-Chem model simulations to estimate wildfire ozone enhancements, and to

compare with those from the Lagrangian and statistical approach as described above. The GEOS-Chem

chemical transport model is driven by the GEOS-5 assimilated meteorological fields from the NASA

Global Modeling and Assimilation Office (GMAO) (http://www.geos-chem.org; v8-02-03) (Bey et al.,

2001). We use a nested version of GEOS-Chem that has 1/2 °×2/3 °horizontal resolution over North

America and adjacent oceans (140 °W-40 °W, 10 °N-70 °N) and 2 °×2.5 ° over the rest of the world. We

reproduce the simulations in Zhang et al. (2014) that also used the wildfire area burned of Yue et al.

(2013) for three-year (2006-2008) ozone simulations over North America. Wildfire ozone enhancements

are computed as differences between the simulation with all emissions turned on and a sensitivity

simulation with only wildfire emissions turned off.


## 3. Model evaluation

We first evaluate our Lagrangian-based Fire Index using its correlation to OC aerosol concentrations, as

previous studies have shown that wildfires are an important source of OC aerosols in the US

Intermountain West in summer (Park et al., 2007; Spraklen et al., 2007). As shown in Figure S3 and

Table S2, the TFI values at each CASTNet site are positively correlated with OC aerosol concentrations

measured at collocated IMPROVE sites (r=0.19-0.44). While the TFI vs. OC correlations are not very

strong reflecting both uncertainties in the FLEXPART retroplumes and influence from other OC aerosol

sources, the correlations are better (p <0.01) than those with areas burned within 10 °×10 ° regions. We

also test the correlations of OC aerosols with Fire Index calculated using trajectory residence time at

lower altitudes or shorter backward time periods, but all show slightly weaker correlations (Table S2).

Table 1 summarizes the predictors included in the MLR models and their performance for each



CASTNet site with more details given in Table S3. The MLR models explain 16%-59% of the

variability in MDA8 ozone concentration among these sites. For the ensemble of 13 CASTNet sites, the

MLR models generally reproduce the ozone measurements ($R^2$=0.60, Figure S3). These coefficients of

determination ($R^2$) are comparable with, or even better than results simulated by Eulerian CTMs (e.g.,

$R^2 = 0.43$ in Zhang et al. (2014), $R^2 = 0.25$ in Emery et al. (2012), $R^2 = 0.48$ in Strode et al. (2015)) that

have limited ability to reproduce the measured ozone variability in the Intermountain West probably due

to the coarse model resolution and complex topography. However, they are lower than results from Jaffe

et al. (2013) or Camalier et al. (2007) that applied the regression models on ozone concentrations at US

urban and low-altitude sites.

Jaffe et al. (2013) analyzed the surface ozone concentrations measured at the SLC urban site in the

western US during June-September 2000-2012, and showed that a MLR model using meteorological

variables as predictors could explain 60% of the MDA8 ozone variation. Here we also applied our MLR

models to MDA8 ozone concentrations at SLC in the summers 1990-2010. We find FI and

meteorological variables can explain 48% of the daily MDA8 ozone variation for summers 1990-2010

(46% if meteorological variables alone are used, and 57% if September data are also considered), which

is a higher value than at most of the CASTNet sites. In addition, as shown in Table 1 the MLR model $R^2$

values for higher-altitude CASTNet sites (> 2000m such as CNT, MEV, PND) are generally lower than



values for lower-altitude sites (such as GLR, CHA and BBE). It appears that the MLR model performs

better for US urban and low-altitude sites than for the CASTNet high-altitude background sites. This is

likely because ozone at the high-altitude CASTNet sites is more affected by regional transport from

both anthropogenic and natural sources such as lightning and stratospheric ozone, and less controlled by

local meteorology relative to ozone at urban or low-altitude sites.

We find that at the CASTNet sites daytime mean RH is generally the most important predictor. In the

low-$NO_x$ background environment, $HO_x$ serves as a strong sink for ozone driving the correlation with

water vapor concentrations (hence RH) (Doherty et al. 2013; Pusede et al., 2015). Fire impacts ($FI_s$ and

$FI_l$) are included for different sites, as it would be expected by their different travel times from the

frequent burning areas to the receptor sites. SqrFI often shows a higher explanatory power than FI,

reflecting nonlinear ozone production from wildfire emissions.

We also acknowledge that the MLR models underestimate high ozone values especially when measured

MDA8 ozone exceeds 70 ppbv (Figure S3). These underestimates, however, are not likely due to model

underestimates of wildfire ozone influences, and may be associated with other factors not included in

the statistical model such as transport from Asia or California or from lightning emissions. As

demonstrated in Figure S4, the MLR model residuals for those high ozone days (MDA8 > 70 ppbv)





have little correlation with TFI, and most of the model underestimates occur when there are small fire

impacts or fires not captured by the FLEXPART retroplumes.

## 4   Results

### 4.1  Consistency and difference with the Eulerian model

It is of particular value to evaluate the MLR wildfire ozone enhancements with those from the Eulerian

approach. We show in Figure 2 such a comparison with the wildfire ozone enhancements estimated by

the GEOS-Chem model in the summer 2007 when there are large wildfire emissions in Idaho. We can

see that the GEOS-Chem model simulates a sharp gradient of wildfire influences with ozone

enhancements greater than 20 ppbv over the Idaho and Montana burning areas, which decrease rapidly

downwind to 0.5-3 ppbv.


To evaluate the MLR wildfire ozone enhancements, we separate the 13 CASTNet sites into three groups

based on their distances to the major burning area in Idaho. As shown in Figure 2, the MLR and

GEOS-Chem estimated wildfire ozone enhancements for all three groups are moderately correlated

($r$=0.34-0.48, statistically significant $p < 0.05$), reflecting some consistency between the two approaches.

There are also considerable differences. We can see that GEOS-Chem simulates up to 40 ppbv wildfire

ozone enhancements for the short-distance sites, much higher than the MLR estimates (mean value of



3.96 ppbv versus 1.85 ppbv). Zhang et al. (2014) has shown that wildfire $NO_x$ emission factor in this

GEOS-Chem simulation is too high by a factor of 3. A sensitivity simulation with a reduced wildfire

$NO_x$ emission factor (from 3.0 g to 1.0 g NO per kg of dry mass burned) would decrease the

GEOS-Chem mean ozone enhancement for the short-distance sites from 3.96 ppbv to 2.06 ppbv. On the

other hand, for the long-distance sites, the GEOS-Chem wildfire ozone enhancements become

substantially lower than MLR (0.77 ppbv versus 1.02 ppbv). As pointed out by Zhang et al. (2014),

GEOS-Chem largely overestimates wildfire ozone influences near the source regions but fails to capture

continued ozone production in wildfire plumes downwind, likely due to the coarse resolution and

inadequate PAN chemistry. The lower GEOS-Chem wildfire ozone estimates at those long-distance sites

may be also attributed to the model difficulty in simulating ozone production from small-scale fires

nearby. The MLR approach appears to show a more reasonable pattern.

## 4.2  Contribution of wildfires to the MDA8 ozone concentration

We use the MLR models to diagnose the influences of wildfires and other meteorological parameters on

MDA8 ozone concentrations at the Intermountain West CASTNet sites. Figure 3 shows the scatter-plots

of observed MDA8 ozone and MLR predicted ozone at four selected sites located in different regions

(GLR, ROM, GRB, and CHA). Also shown are the boxplots of MLR wildfire ozone enhancements and

MLR no wildfire ozone as defined by Equation (4) and (5), respectively. The MLR models generally



reproduce the measurements except for high ozone values as we have discussed above. For all the

CASTNet sites, the MLR no wildfire ozone explains most of the measured MDA8 variability

($R^2$=0.10-0.58) compared to MLR wildfire ozone enhancements ($R^2$=0.02-0.12). However, wildfire

ozone enhancements increase as measured MDA8 ozone concentrations increase, reflecting higher

wildfire impacts on the high-ozone events. We can see in a few cases wildfire ozone enhancements

reach 10-20 ppbv, causing measured MDA8 ozone to approach the ozone quality standard of 70 ppbv.

Another test to separate wildfire ozone influences from meteorological impacts follows Jaffe et al.

(2008) who showed ozone concentrations in high-fire years were distinctly greater than those in

low-fire years at the same temperature ranges. Here we extend their approach to other meteorological

parameters and to the whole 22-year records. Figure 4 shows the relationships between MDA8 ozone

concentrations and meteorological parameters (daytime temperature, wind speed, RH, and solar

radiation flux) measured at a Chiricahua National Monument, Arizona (CHA). We compare measured

MDA8 ozone concentrations with high versus low wildfire impacts (upper 33% versus lower 33% of

the TFI values). Meteorological variations have some impacts on both wildfire activities and MDA8

ozone levels. High wildfire events are prone to occur with high temperature and solar radiation, low RH

and wind speed, as indicated by the number of upper 33% versus lower 33% TFI occurrences in each

increment of meteorological parameters. Ozone concentrations generally increase with increasing





temperature and decreasing RH. We can also see significant differences ($p< 0.05$) in the MDA8 ozone

concentrations between the upper and lower TFI values for most of the meteorological increments. For

instance, in the 26-28 ℃ temperature bin, the mean MDA8 ozone for the upper 33% TFI is about 8 ppbv

higher than that for the lower 33% TFI. This confirms impacts of wildfires on ozone that are

independent from meteorological variables.

### 4.3  Wildfire influences on the ozone interannual variability and trend

Application of the MLR models to the summers 1989-2010 ozone measurements allows us to quantify

wildfire influences on the long-term ozone variability and trend. We show in Figure 5 time series of

summer mean measured and MLR predicted MDA8 ozone concentration for the Intermountain West

regional average, as well as for three individual sites (GLR, YEL, and GRC) in the 22 years

(1989-2010). The MLR models show good agreements with measurements with correlation coefficients

of 0.85 for the regional average and 0.52-0.92 for individual sites, but underestimate the measured

interannual variability. Figure 5 also shows the summer mean MLR with and without the wildfire ozone,

along with the difference between the two. The interannual variability of surface ozone over the region

appears to be more controlled by the interannual variations of meteorological parameters, and hence the

climate variability, as we can see that even without wildfire influences, the remaining meteorological

parameters used in the MLR models still predict most of the interannual variability (MLR no wildfire



ozone vs. MLR ozone $r = 0.87$-$0.99$ among individual sites).

Wildfires contribute 0.3-1.5 ppbv to the summer mean surface MDA8 ozone averaged over the

Intermountain West CASTNet sites. In the high-fire activity years such as 2003 and 2007, the mean

wildfire ozone enhancements can reach 3.5 ppbv at the individual sites, e.g., MEV. The interannual

variability of wildfire ozone enhancements is strongly correlated with that of the MLR total ozone ($r =$

0.89 for the regional averages and 0.48-0.87 for individual sites). As we can see here, the

wildfire-driven interannual variability is much weaker than what can be explained by meteorological

parameters. We thus suggest that some of the strong correlation between summer mean surface ozone

concentrations and wildfire activities reflects their common relationships with meteorological

parameters such as RH and temperature at the interannual scale. However we should acknowledge that

ozone production in wildfires varies significantly (Jaffe et al., 2012), and the statistical models we use

here can still underestimate the interannual variations of wildfire influences. Better resolving the causes

of variations in wildfire ozone production will help us understand the source for interannual variations

in ozone.

We further calculate the linear trends of surface ozone in the summers 1989-2010. Figure 6 summarizes

the results at three percentile ranges: 93-97[th], 48-52[th], 3-7[th] percentiles at the Intermountain West



CASTNet sites. We also show the separated trends for the earlier (1989-1999) and later (2000-2010)

periods following Strode et al. (2015) who suggested different trends in surface ozone for the two

periods. Regional averaged summer MDA8 ozone concentrations in the Intermountain West show

increasing but statistically insignificant trends of $0.14\pm0.21$ (p=0.22), $0.19\pm0.21$ (p=0.08), and

$0.18\pm0.20$(p=0.09) ppbv year$^{-1}$ at the 93-97$^{th}$, 48-52$^{th}$, and 3-7$^{th}$ percentiles, respectively, in 1989-2010.

Statistically significant (p<0.05) increasing trends are found at the YEL ($0.42\pm0.30$ ppbv year$^{-1}$) and

ROM ($0.43\pm0.39$ ppbv year$^{-1}$) sites at the median percentiles. These increasing trends primarily

occurred in the earlier period (1989-1999), while nearly all sites show decreasing ozone trends during

2000-2010. Strode et al. (2015) attributed the earlier increasing trends to meteorological variations and

the later decreasing trends to domestic emission controls. Our results are consistent with previous

studies of Cooper et al. (2012) and Strode et al., (2015) who analyzed the ozone trends using the same

CASTNet measurements but using the metric of daytime ozone concentration.

Also shown in Figure 6 are the corresponding ozone trends contributed by wildfires as estimated by the

MLR models. A distinct feature is that the trends of wildfire ozone enhancements are relatively small

but generally in the same directions as the observed ozone trends. This feature can also result from

meteorological variations that modulate surface ozone concentrations and wildfires in similar directions.

Most of the sites show increasing wildfire ozone in the first 11 years (1989-1999), and switch to



decreases in the next 11 years (2000-2010), but only a few of them are statistically significant. Wildfire

ozone enhancements averaged over the Intermountain West CASTNet sites increase at rates of $0.02\pm$

0.05 ppbv year$^{-1}$ (p=0.48), $0.02\pm0.05$ ppbv year$^{-1}$ (p=0.38), and $0.03\pm0.03$ ppbv year$^{-1}$ (p<0.05) at the

93-97$^{th}$, 48-52$^{th}$, and 3-7$^{th}$ percentile ranges, respectively, in the summers 1989-2010. These values

account for about 15% of the observed ozone trends at the same CASTNet sites, representing small but

important ozone influences from wildfires.

## 4.4  Wildfire influences on ozone exceedance days

As the ozone air quality standard becomes stricter, it is important to quantify the number of ozone

exceedances caused partly by uncontrollable sources, such as wildfires. We show in Figure 7 the mean

number of days with measured MDA8 ozone concentrations exceeding 75 ppbv, 70ppbv, and 65 ppbv

averaged over the 13 Intermountain West CASTNet sites in the summers 1989-2010. Also shown is the

corresponding number of exceedances that would be present in the absence of wildfires (estimated as

measured ozone minus the MLR wildfire ozone). In the years with poor air quality conditions such as

2002 and 2003, there were more than 20 days when MDA8 ozone exceeds 65 ppbv accounting for 22%

of the summer days, and about 8 days with MDA8 exceeding 70 ppbv, the current ozone air quality

standard. However, if there were no wildfire emissions, the frequency of ozone exceedance days would

significantly decrease. For the total exceedance days at the 13 sites in this period, the number with



MDA8 above 65 ppbv (above 70 ppbv) would decrease by 28% to 1509 days (by 31% to 474 days).

This reduction is particularly important in high fire years such as 2002-2003 and 2005-2007 when one

third to half of the exceedances would not occur without the fires.

## 5    Conclusions

Simulating the complexity in wildfires emission and chemistry requires running Eulerian models at very

fine resolution (Jiang et al., 2012; Jaffe et al., 2013; Zhang et al., 2014), which presents challenges for

assessing long-term wildfire ozone enhancement using those models. In this study, we have applied a

new approach based on a Lagrangian particle dispersion model (FLEXPART) and statistical models to

quantify the wildfire influences on the ozone daily and interannual variability, trends, and exceedance

days over the US Intermountain West in the summers 1989-2010. The recent implementation of a more

stringent ozone standard (70 ppbv) in the United States also motivates the need to better understand

contributions and variations of natural ozone sources such as wildfires.

We introduce a Fire Index (FI), a measure of wildfires' impact at a receptor site, by using 5-day

FLEXPART retroplumes (plumes of back-trajectory particles) combined with a daily high-resolution

wildfire area burned dataset in the Western US. The FI values are computed for each ozone

measurement day in the summers 1989-2010 for the ensemble of 13 CASTNet sites and an urban site



(SLC) over the US Intermountain West. We then develop statistical MLR models that estimate MDA8

ozone concentrations at each site as a function of FI and various meteorological variables. We show that

the MLR models explain 60% of the variability of MDA8 ozone over the US Intermountain West

(16%-59% at individual sites), which is comparable with results from current Eulerian CTMs ($R^2$ =

0.25-0.48 as reported in recent studies (Zhang et al., 2011; 2014; Emery et al., 2012; Strode et al.,

2015)).

The MLR models allow us to diagnose the MDA8 ozone enhancements from wildfires as well as ozone

controlled by meteorological variables. We compare wildfire ozone enhancements estimated by the

MLR models with those from the GEOS-Chem CTM for summer 2007. While some consistency is

found, the two methods show rather different patterns. The MLR method appears to better capture

wildfire ozone influences at larger distances downwind of the fires or ozone produced from small-scale

fires. We find that wildfire ozone enhancements estimated by the MLR models occasionally reach

10-20 ppbv at the Intermountain West CASTNet sites, and they tend to increase as measured ozone

concentrations increase, reflecting higher wildfire impacts on the high-ozone days. Meteorological

variations also show distinct impacts on both wildfire activities and MDA8 ozone concentrations. High

wildfire events and high ozone days are often associated with high temperatures and strong solar

radiation, and low RH and wind speed.



We find wildfires increase the summer mean MDA8 ozone concentrations by 0.3-1.5 ppbv averaged

over the Intermountain West CASTNet sites during 1989-2010. While the interannual variability of

summer mean wildfire ozone enhancements is strongly correlated with that of the MLR total ozone, the

wildfire-driven interannual variability is much weaker than the ozone variability that can be explained

by meteorological parameters. We suggest that the strong interannual correlation between summer mean

ozone concentrations and wildfire activities can be partly driven by their common relationships with

meteorological parameters such as RH and temperature. These common relationships may also be

responsible for the synchronous trends of summer mean surface MDA8 ozone concentrations and

wildfire ozone enhancements for either the 1989-2010 period or two separated 11-year periods

(1989-1999 vs. 2000-2010). Wildfires contribute about 15% to the observed ozone trends at the

Intermountain West CASTNet sites.

Wildfires thus present an important source affecting surface ozone air quality in the US Intermountain

West. Despite small enhancements when averaged seasonally or regionally, they have notable impact on

the occurrence of ozone exceedances, reflecting the small-scale and episodic nature of wildfire

emissions. We show that about one third of the summer days (1989-2010) with MDA8 ozone exceeding

70 ppbv would not occur in the absence of wildfires. While we have shown that our Lagrangian and



statistical approach provides a quantitative estimate of ozone enhancements from wildfires, and can be

applied to analyze long-term ozone records, there are still considerable uncertainties in this approach

from both the FLEXPART calculation and the MLR models as discussed in the text. The approach also

does not consider the complexity in fire emissions and cannot probe into the physical and chemical

processes in the fire plumes. To address this issue would require more detailed fire plume

measurements and finer-scale modeling approaches, such as imbedding a plume-in-grid model to

CTMs.

## Acknowledgements

This work was supported by China's National Basic Research Program (2014CB441303), and by the

National Natural Science Foundation of China (41475112).


**4 Figures and 3 Tables are included in the supplement related to this article.**

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



## Figures and Tables

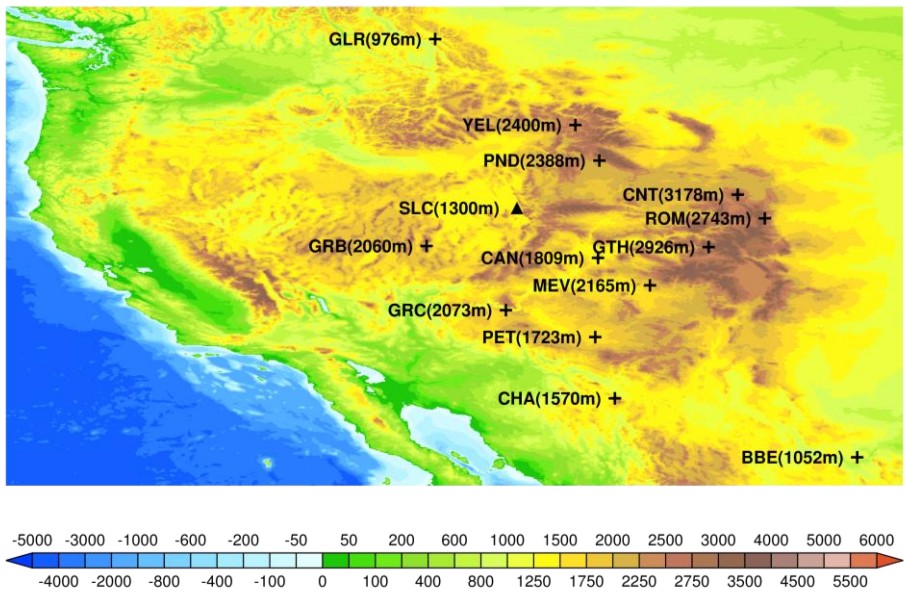

**Figure 1**. 13 CASTNet ozone monitoring sites (Table 1, black pluses) in the US Intermountain West

used in this study. Also shown is SLC (Salt Lake City, Utah) urban site (filled triangle). Altitudes of the

sites are also labeled. The underlying figure shows terrain elevations of the western US.





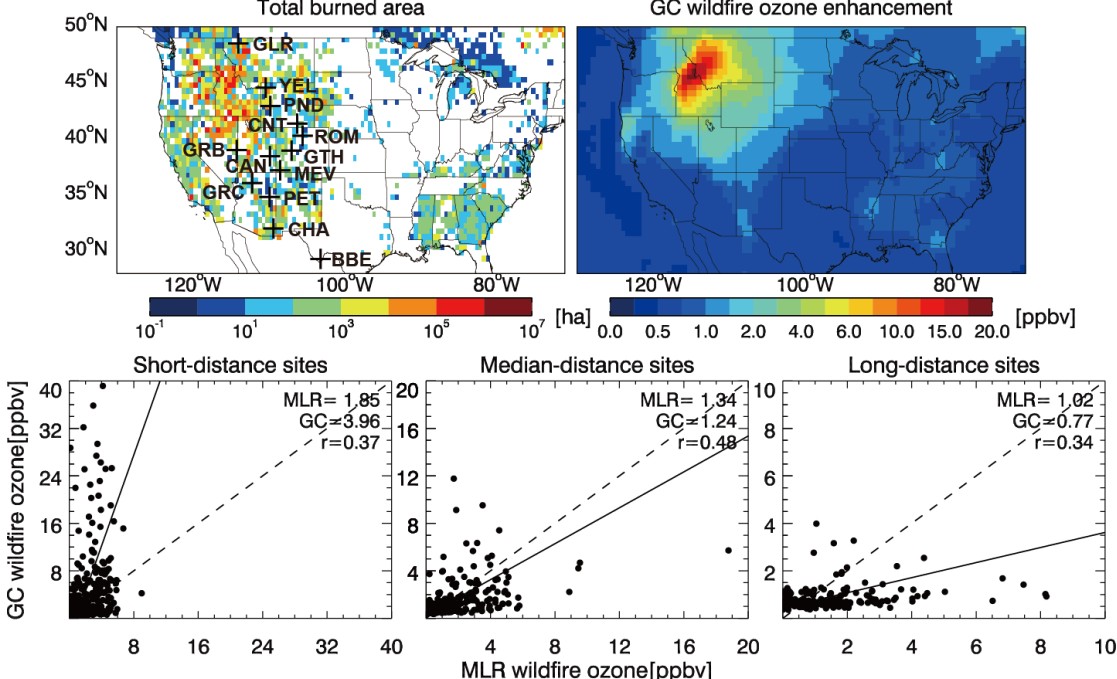

**Figure 2**. Wildfire ozone enhancements over the Intermountain West US in summer 2007. Top panels show the total burned area (upper-left panel) and seasonal mean wildfire ozone enhancements computed by the GEOS-Chem simulation (upper-right panel). Wildfire ozone enhancements computed by the MLR models are compared with those by the GEOS-Chem simulation. The comparisons are separated
by their distances to the location with the maximum fire emission in Idaho: short-distance sites (bottom-left; GLR, YEL, PND, and GRB), median-distance sites (bottom-middle; CNT, ROM, GTH, CAN, MEV, and GRC), and long-distance sites (bottom-right; PET, CHA, and BBE). Mean wildfire ozone enhancements, correlation coefficients (r), reduced-major-axis regression lines (solid) and 1:1 lines (dashed) are shown inset.





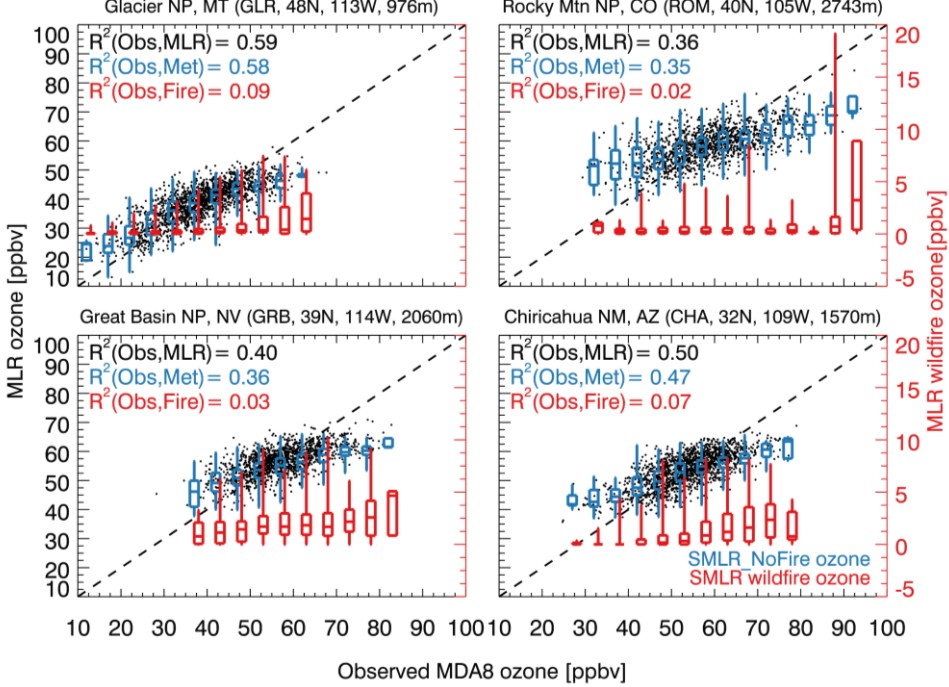

**Figure 3**. Scatter-plots of observed versus MLR predicted MDA8 ozone concentrations at 4 selected CASTNet sites for the summers 1989-2010. Also shown are the box-and-whisker plots (minimum, 25th, 50th, 75th percentile, and maximum) of ozone without wildfire influences (blue) and wildfire ozone enhancements (red) for 5-ppbv bins of observed ozone concentrations; both are computed by the SMLR model as described in the text. The 1:1 line (dashed line) and the coefficient of determination ($R^2$) are shown inset.



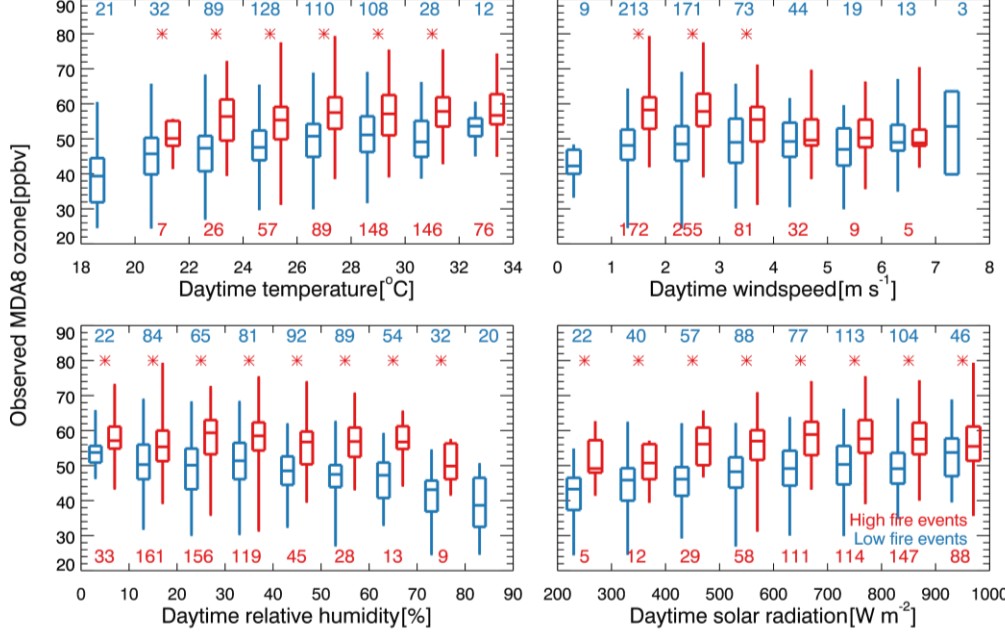

**Figure 4**. Box-and-whisker plots (minimum, 25th, 50th, 75th percentile, and maximum) of observed MDA8 ozone concentrations for bins of observed daytime meteorological parameters at CHA site: temperature (upper-left), wind speed (upper-right), relative humidity (bottom-left), and solar radiation flux (bottom-right). MDA8 ozone concentrations are divided by high (TFI at top 33%, red) and low (TFI at lower 33%, blue) fire events with the number of occurrences in each bin shown inset. Significant difference ($p < 0.05$) is marked by asterisks.



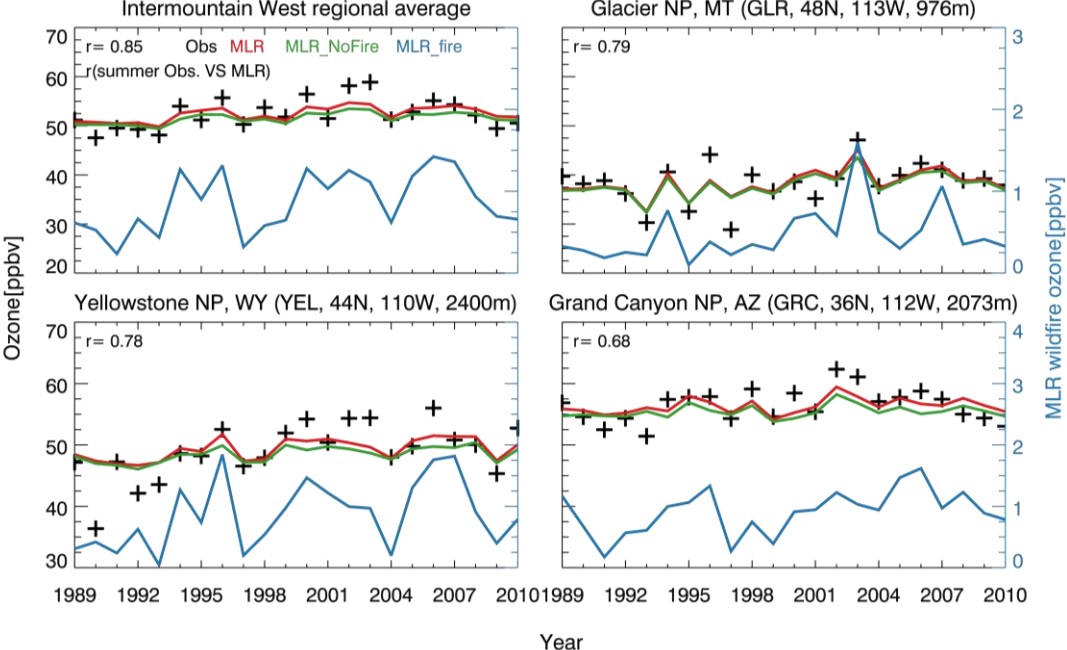

**Figure 5**. Time series of summer mean MDA8 ozone concentrations for the regional averages of 8
CASTNet sites with complete 22-year measurements as well as 3 individual sites. Measurements (black
pluses) are compared to the SMLR model results (red line). Also shown are the summer mean SMLR no
wildfire ozone (green line) and SMLR wildfire ozone (blue line, right axis). The correlations between
measured and SMLR summer means are shown inset.




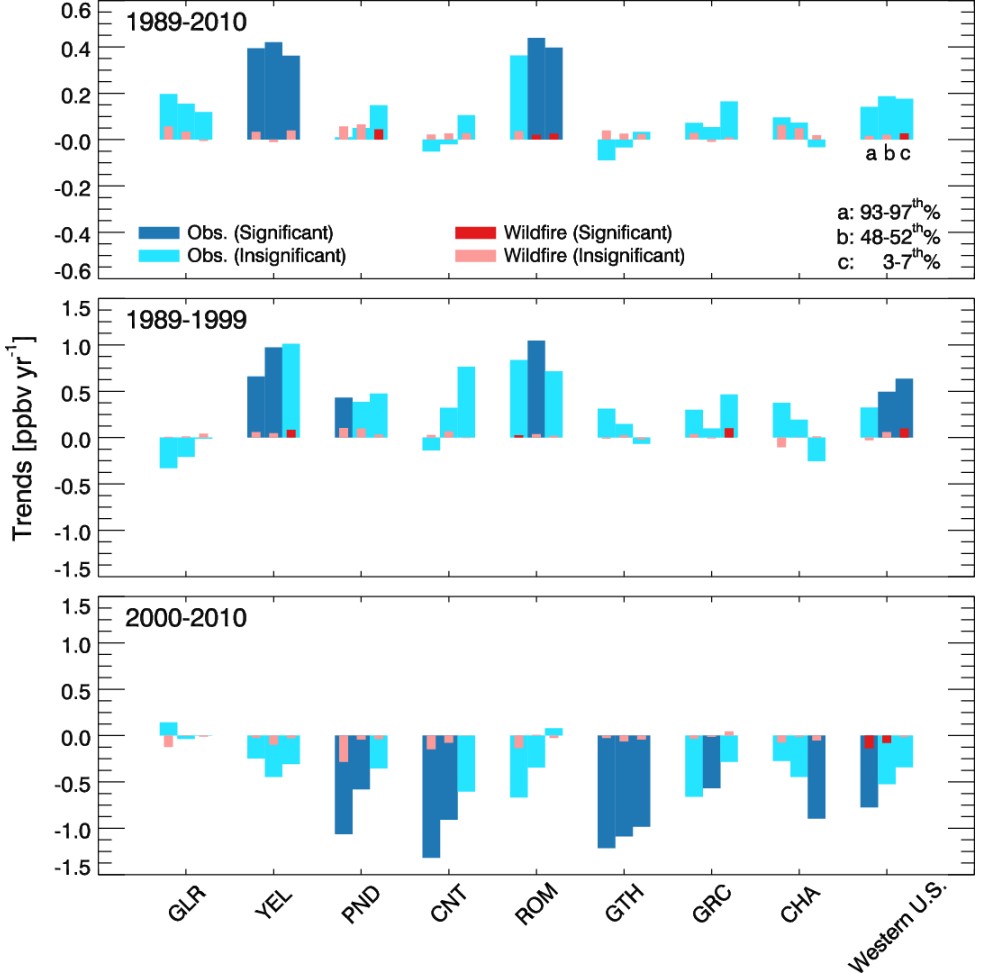

**Figure 6**. Linear trends of summer mean MDA8 ozone concentrations (blue bars) for 1989-2010 (top

panel), 1989-1999 (middle panel) and 2000-2010 (bottom panel) at 8 CASTNet sites and for the

Intermountain West regional averages for the (a) 93th-97th, (b) 48th-52th and (c) 3-7th percentile ranges.

Also shown are the trends contributed by wildfire ozone enhancements (red bars) as computed by the

SMLR models. Statistically significant trends (p < 0.05) are emphasized in dark color.



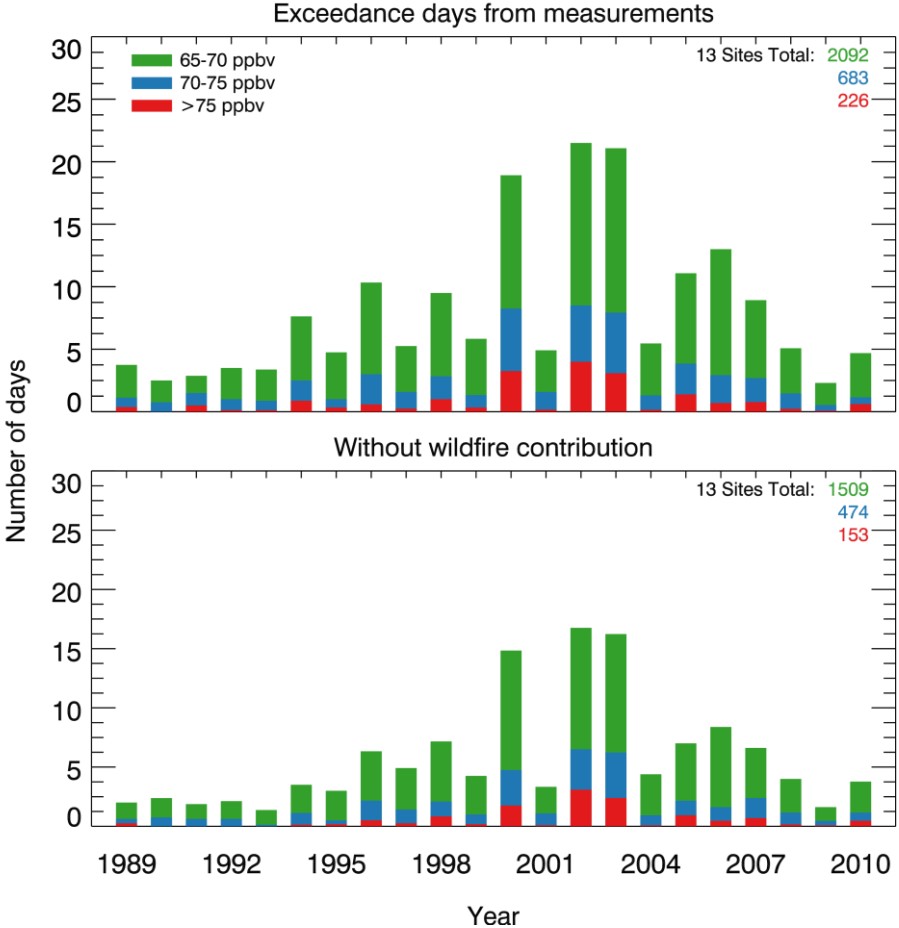

**Figure 7**. Mean number of days with MDA8 ozone concentrations exceeding the thresholds of 65, 70 and 75 ppbv averaged over the 13 CASTNet sites in the Intermountain West for the summers 1989-2010. The top panel shows the exceedances computed from the measurements, and the bottom panel shows results that would be presented in the absence of wildfires (measurements minus the SMLR estimated

wildfire ozone enhancements).

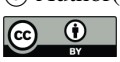

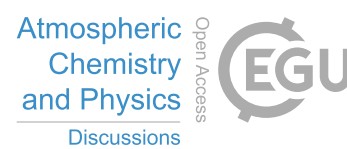
**Table 1.** Multiple linear regression (MLR) models for summer MDA8 ozone at 13 Intermountain West CASTNet sites[a]

| Sites[b] | $R^2$ (N) | Variables included in the MLR model[c] |
|---|---|---|
| **Glacier NP, MT** (GLR, 48N, 113W, 976m) | 0.59 (1809) | RH, WSPsurf, SRAD, U, V, Ome, SH, HGT, T, dT, SH, SqrFI_l, SqrFI_s, |
| **Yellowstone NP, WY** (YEL, 44N, 110W, 2400m) | 0.35 (1611) | RH, WSPsurf, Tsurf, SRAD, U, V,WSP, OME, HGT, T, dT, SH, SqrFI_l, SqrFI_s, FI_l, |
| **Pinedale, WY** (PND, 42N, 109W, 2388m) | 0.28 (1888) | RH, WSPsurf, Tsurf, SRAD, U, V, WSP, Ome, HGT, T, SH, SqrFI_l, SqrFI_s, FI_s, |
| **Centennial, WY** (CNT, 41N, 106W, 3178m) | 0.19 (1925) | RH, U, WSP, HGT, T, SH, SqrFI_l, SqrFI_s, FI_s, |
| **Rocky Mtn NP, CO** (ROM, 40N, 105W, 2743m) | 0.36 (1367) | RH, WSPsurf, Tsurf, SRAD, PRCP, U, Ome, T, SH, FI_s, SqrFI_l, SqrFI_s, |
| **Gothic, CO** (GTH, 38N, 106W, 2926m) | 0.29 (1906) | RH, WSPsurf, U, V, WSP, Ome, HGT, T, dT, SH, SqrFI_l, FI_l, |
| **Mesa Verde NP, CO** (MEV, 37N, 108W, 2165m) | 0.23 (1321) | RH, WSPsurf, Tsurf, SRAD, U, V, T, dT, SqrFI_l, SqrFI_s, |
| **Great Basin NP, NV** (GRB, 39N, 114W, 2060m) | 0.40 (1360) | WSPsurf, Tsurf, SRAD, U, WSP,Ome, SH, Ome, HGT, SH, SqrFI_l, SqrFI_s, FI_s, |
| **Canyonlands NP, UT** (CAN, 38N, 109W, 1809m) | 0.16 (1379) | RH, WSPsurf, Tsurf, V, Ome, T, FI_l, SqrFI_l, SqrFI_s, |
| **Grand Canyon NP, AZ** (GRC, 36N, 112W, 2073m) | 0.34 (1912) | RH, WSPsurf, SRAD, PRCP, U, V, WSP, Ome, HGT, T, SH, SqrFI_l, FI_l, |
| **Petrified Forest, AZ** (PET, 34N, 109W, 1723m) | 0.43 (654) | RH, SRAD, V, WSP, Ome, HGT, T, dT, SH, SqrFI_l |
| **Chiricahua NM, AZ** (CHA, 32N, 109W, 1570m) | 0.50 (1754) | RH, SRAD, PBLH, U, V, WSP, HGT, T, dT, SH, SqrFI_l, FI_l, |
| **Big Bend NP, TX** (BBE, 29N, 103W, 1052m) | 0.46 (1196) | RH, WSPsurf, SRAD, U, V, WSP, HGT, T, SqrFI_l, SqrFI_s, FI_l, FI_s |

[a] Coefficients of determination ($R^2$), sample numbers (N), and variables included in the MLR models.

[b] NP = National Park, NM = National Monument, MT = Montana, WY = Wyoming, CO = Colorado, NV= Nevada, UT = Utah, AZ = Arizona, TX = Texas.

[c] Fire Index (FI, FI_s), square root of FI (SqrFI_l, SqrFI_s), and meteorological parameters including (1) surface measurements: daytime (10:00-17:00 local time) mean temperature (Tsurf), wind speed (WSPsurf), relative humidity(RH), and solar radiation flux (SRAD); (2) gridded daily precipitation (PRCP); (3) NCEP data at 850/700/500 hPa pressure levels: daily maximum planetary boundary layer height (PBLH), daily mean zonal wind speed (U), meridional wind speed (V), horizontal wind speed (WSP), temperature(T), geopotential height (HGT), vertical velocity (Ome), specific humidity (SH), and temperature at 1000hPa minus that at 850 hPa (dT).

Please refer to Table S1 and S3 for details on the parameters and MLR models.
