# Peer review of "Wildfire influences on the variability and trend of summer surface ozone in the mountainous western United States"

_Atmospheric Chemistry and Physics, 2016_

## Referee Comment (RC1) · Anonymous Referee #1 · 2 Sep 2016

Overview: The paper presents a new approach to examining the influence of wildfire smoke on ozone mixing ratios at remote/rural monitoring sites in the U.S. intermountain west. Overall the paper is well written and suitable for publication in ACP. I recommend that the authors consider the following ideas in revising the manuscript.

1) Line 285: The sentence staring with "These underestimates" requires substantially more justification/analysis/references.

2) Line 315: There are many reasons that a model like GEOS-Chem will not adequately represent the role of fires. The standard versions of GEOS-Chem do not emit short lived VOCs, and the emission factors for NOx emissions from fires are quite variable in reality. The model also adds all the emissions within the boundary layer. The authors

clearly recognize this because they use a 5km cut off for the FLEXPART analysis, and are certainly aware of recent work by Val Martin et al. (e.g. 2010) with respect to plume heights over North America. This should be discussed in depth or omitted. A reference to Zhang et al., (2014) is inadequate.

3) Why does this paper narrowly focus on the intermountain west? This region has many wildfires, but the smoke travels and the impact on ozone may be larger downwind (see Brey and Fischer, 2016). S. Brey and E.V. Fischer (2016), Smoke in the City: How often and where does smoke impact summertime ozone in the United States, Environ. Sci. Tech.,DOI:10.1021/acs.est.5b05218.

4) I have two questions with respect to Figure 7 (and the associated discussion). First, is it appropriate to use the entire range of 1989-2010 to look at the number of exceedance days. There have been trends in ozone during this time. Second, and more importantly, would it be more appropriate to view the exceedance days as a percentage of the total, rather than as a count. Yes, there will be more exceedance days as we lower (tighten) the ozone standard, all things held the same. However, do we have a way to determine if the relative importance of fires will increase?

5) Finally, I think all the SI materials should be moved into the main paper. There are very important figures in the SI materials, and I had to refer to them to follow the paper. Without them in the main manuscript, it would be easy to overlook the fact that the MLR really does not do a good job reproducing the highest ozone days. This is an important point in considering the value of this analysis.

———————————————

---

## Referee Comment (RC2) · Anonymous Referee #2 · 26 Sep 2016

This paper uses back trajectories form the Lagrangian particle dispersion model FLEX-PART and estimated fire emissions for the years 1989-2010 to define a Fire Index for 13 CASTNet sites in the Intermountain West. This fire index and various meteorological parameters are used as predictors in a multi-linear regression (MLR) model that predicts daily MDA8 O3 at these sites. The estimated impact of the fire index terms in the model is then used to determine the influence of wildfires on the MDA8 O3, and this estimate is compared to estimated of the Eulerian chemical transport model GEOS-Chem. The authors find that wildfires enhance the summer mean MDA8 O3 by 0.3-1.5 ppbv, with episodic daily increases of 10-20 ppbv at individual sites. They find that GEOS-Chem tends to over-predict the near source formation of O3 and underpre-

dict the downwind formation, consistent with previous Eulerian model studies. Finally, they find that the influence of wildfires is especially important on high O3 days, where 31% of the days with MDA8 O3 over 70 ppbv would not have occurred in the absence of wildfires.

This is a well-done, innovative study and a well-written manuscript. The development of the fire index and the MLR both help to understand the complex influence of fire emissions on O3 in the intermountain west and to expose errors in Eulerian models of this process. The methodology is generally sound and the results are consistent with our understanding of fire chemistry. While I have a few concerns that I would like to see addressed before publication, as detailed below, in general this is a very nice study that should be published.

Major Concerns:

I have concerns with two of the conclusions of the paper:

1. The abstract states (P2, L32-33) that wildfires contribute 15% of the measured increasing but statistically insignificant trend in MDA8 O3, and this is also stated in the conclusions section (P26, L461-462). However, as neither trend is statistically significant, I disagree with including the 15% value as a major conclusion of the paper, where it might be erroneously quoted without proper context. Thus I recommend that the abstract and conclusion statements be removed from the paper, but the discussion in Section 4.3 remain, as the trend results are given proper context there. 2. P20, L357-360 states that the interannual variability in MDA8 O3 appears to be more controlled by interannual variations of the meteorological parameters, as the meteorological variables can account for "most" of the interannual variability in the MDA8 O3, even without fires. I do not think this conclusion is adequately supported by the presented evidence. The fact that the MLR for the met parameters has roughly the same interannual variability as the measurements could be just a statistical artifact of the MLR procedure, with the interannual variability incorrectly accounted for by the meteorological predictors. The conclusion would be more convincing if evidence were presented of the interannual variability of specific meteorological parameters (Temperature, RH, etc.), and if the highs and lows in the summer means of these raw variables were consistent with the highs and lows in MDA8 O3.

Minor Concerns: P7, L122: Please add the latitude-longitude or ID number for the Sal Lake City site you are using.

P9, L156: 250,000 is a huge number of particles to track, and is probably overkill. Usually 500 particles per receptor (time step and location) is sufficient. How many time steps are there each day, and how many particles are released in each time step?

P9, L157-158: Is this e-folding time supposed to account for the deposition of smoke along the path? How is this done – is a number on each particle decreased, or do some of the particles actually disappear during the simulation?

P10, L169: My understanding is that MISR observations suggest that plumes go above the boundary layer 20-25% of the time, so "often" seems a little vague and misleading.

P10, L170: It is true that 5 km and 5 days generally gave the best correlation, but the change in the fit wasn't very significant compared to 2 km (PBL height) and 5 days.

P10, L175: I'd like to see an equation for variable tr(i,j) as well, that shows how the residence time for a single layer is calculated and how the layers are integrated vertically.

P11, L189: Since Table S1 defines the variables, I think it should be moved to the main paper.

P11, L198: I understand the choice of MLR limits what you can do to look at non-linearity, but why did you choose the square root of the index instead of, say, the square of the index?

P11, L202: You should briefly discuss how the model doesn't include interaction terms between the predictors, and the effect this might have on the model performance.

P14, L236: Do you mean this is just a reanalysis of the Zhang et al. (2014) output, or did you rerun the simulations? You note later that the NOx emissions in this simulation are too high – why didn't you use the lower value here?

P14, L250: I think you mean "all except for GRC" show weaker correlations, or ther is an error in Table S2.

P15, L267: Can you explain why you get poorer correlations for Salt Lake City than in the Jaffe et al. (2013) study? What does this imply for your other results?

P15, L270: I think this dependence of the performance on altitude makes sense, but a scatter plot of R2 versus site altitude in the supplement would help to prove it.

P15-16, L287-L290: Since you include the fire index as a predictor, the fact that the residuals don't correlate with TFI just shows that the MLR procedure is working as expected, right? The second clause of this sentence, that underestimates occur even in the absence of fires, seems like more convincing evidence to me.

P21, L367-L371: I didn't understand what you were trying to say here – please elaborate or rephrase?

P21, L377: Can you please explain why you chose these percentile ranges?

P24, L420-422: I'd suggest cutting this sentence, as the context of the study is already established in the introduction and this statement is incomplete – Eulerian model errors are not just about resolution, but about errors in amount, location, and timing of biomass burned, error in emission speciation, errors in chemistry, numerical diffusion errors, etc.

P25, L435: Make clear how this average R2 is calculated.

Typos and Wording Suggestions: P7, L112-L113: I'd suggest making this a single list: "ozone, organic carbon (OC) aerosols, meteorological parameters, and wildfire area burned data"

P7, L115: Expand CASTNet acronym and provide a little more descriptions than just the website.

P11, L189: Period should go after the parentheses, not before.

P14, L244: Figure S2, not S3.

P14, L247: Need a comma after "strong"

P16, L280: "as would be expected" delete "it"

P21, L364: Say "summer mean" to be as clear as possible.

P22, L389: Don't need comma after "Strode et al."

P23, L411-412: I suggest putting parentheses around the phrase "accounting for 22% of the summer days"

P25, L437-438: I suggest cutting everything after the $R^2$ value - these references are already discussed in the main text and do not need to be repeated here.

P25, L442-443: I don't see much consistency at all between the MLR and GEOS-Chem predictions, so you need to make clearer what consistencies you see.

P27, L474: "model in" instead of "model to"

P33, L677: Add unit '(m)' of terrain elevations to caption, as it is not on the figure color bar.

P34, L689: "those from the GEOS-Chem" instead of "those by the GEOS-Chem"

P38, Figure 6: The wildfire trend values are very hard to see – maybe plot on a secondary y axis? In addition, since the trends are generally not statistically significant perhaps this could be moved to the supplement?

P38, L734: remove "S" from "SMLR" for consistency with the rest of the paper.

P40, L741: Need a space between "relative humidity" and "(RH)"

Figure S1, Caption, L3 and 7: "residence time" not "resident time"

Table S1, Footnotes, L37: Should say "m (PBLH, HGT)", delete the rest.

Table S1, Footnotes, L38: change to "mean represents the average"

Table S3, Footnote c: Put the explanation for the bold text in the figure caption, not the footnote.

---

## Author Comment (AC2) · 6 Nov 2016

**Comment:** This paper uses back trajectories from the Lagrangian particle dispersion model FLEXPART and estimated fire emissions for the years 1989-2010 to define a Fire Index for 13 CASTNet sites in the Intermountain West. This fire index and various meteorological parameters are used as predictors in a multi-linear regression (MLR) model that predicts daily MDA8 O3 at these sites. The estimated impact of the fire index terms in the model is then used to determine the influence of wildfires on the MDA8 O3, and this estimate is compared to estimated of the Eulerian chemical transport model GEOS-Chem. The authors find that wildfires enhance the summer mean MDA8 O3 by 0.3-1.5 ppbv, with episodic daily increases of 10-20 ppbv at individual sites.

[Figure]

They find that GEOS-Chem tends to over-predict the near source formation of O3 and under-predict the downwind formation, consistent with previous Eulerian model studies. Finally, they find that the influence of wildfires is especially important on high O3 days, where 31% of the days with MDA8 O3 over 70 ppbv would not have occurred in the absence of wildfires.

This is a well-done, innovative study and a well-written manuscript. The development of the fire index and the MLR both help to understand the complex influence of fire emissions on O3 in the intermountain west and to expose errors in Eulerian models of this process. The methodology is generally sound and the results are consistent with our understanding of fire chemistry. While I have a few concerns that I would like to see addressed before publication, as detailed below, in general this is a very nice study that should be published.

**Response:** We thank the reviewer for the valuable comments. All of them have been implemented in the revised manuscript. Please see our itemized responses below.

**Comment:** Major Concerns: I have concerns with two of the conclusions of the paper: 1. The abstract states (P2, L32-33) that wildfires contribute 15% of the measured increasing but statistically insignificant trend in MDA8 O3, and this is also stated in the conclusions section (P26, L461-462). However, as neither trend is statistically significant, I disagree with including the 15% value as a major conclusion of the paper, where it might be erroneously quoted without proper context. Thus I recommend that the abstract and conclusion statements be removed from the paper, but the discussion in Section 4.3 remain, as the trend results are given proper context there.

**Response:** We agree that considering the complexity of wildfire impacts and uncertainties in the trends, 15% is not well constrained. We now remove the statements from the abstract and conclusion, while keeping them in the discussion section.

**Comment:** 2. P20, L357-360 states that the interannual variability in MDA8 O3 appears to be more controlled by interannual variations of the meteorological parameters, as the meteorological variables can account for "most" of the interannual variability in the MDA8 O3, even without fires. I do not think this conclusion is adequately supported by the presented evidence. The fact that the MLR for the met parameters has roughly the same interannual variability as the measurements could be just a statistical artifact of the MLR procedure, with the interannual variability incorrectly accounted for by the meteorological predictors. The conclusion would be more convincing if evidence were presented of the interannual variability of specific meteorological parameters (Temperature, RH, etc.), and if the highs and lows in the summer means of these raw variables were consistent with the highs and lows in MDA8 O3.

**Response:** Thanks for pointing it out. To support this conclusion, we now present such a figure in supplement (Figure S4) comparing the interannual variability of relative humidity and temperature with MDA8 ozone.

We also state in the text: "This is further supported by the strong interannual correlations between summer mean MDA8 ozone and meteorological parameters such as daytime mean RH and surface temperature at individual sites and for the regional averages (r = -0.69 for RH, r = 0.48 for temperature), as shown in Figure S4."

**Comment:** Minor Concerns: P7, L122: Please add the latitude-longitude or ID number for the SalLake City site you are using.

**Response:** We now state here "we use hourly ozone measurements from 1990-2010 at the Salt Lake City (SLC, 40.6N, 111.9W, 1300m) urban site (data available at https://www3.epa.gov/airdata/) for comparison with the CASTNet background sites and the previous work of Jaffe et al. (2013)."

**Comment:** P9, L156: 250,000 is a huge number of particles to track, and is

probably overkill. Usually 500 particles per receptor (time step and location) is sufficient. How many time steps are there each day, and how many particles are released in each time step?

**Response:** The particle number is selected to ensure that model calculated retroplumes are statistically robust. It is in the middle of two previous studies using the FLEXPART model: 40000 in Cooper et al. (2010) and 1 million in Stohl et al. (2012).

To address this comment, we now state in the text: "For each day at a receptor site, FLEXPART was run in backward mode, with 250,000 particles released at the site location at a constant hourly rate (˜10k particles per hour) during the first 24 hours. Previous studies have used the particle sizes of 40000 (Cooper et al., 2010) and 1 million (Stohl et al., 2012) represent a retroplume."

Reference:

Cooer et al., Increasing springtime ozone mixing ratios in the free troposphere over western North America, Nature, 463, 344-348, doi: 10.1038/nature08708, 2010.

Stohl et al., Xenon-133 and caesium-137 releases into the atmosphere from the Fukushima Dai-ichi nuclear power plant: determination of the source term, atmospheric dispersion, and deposition, Atmos. Chem. Phys., 12, 2313-2343, doi: 10.5194/acp-12-2313-2012, 2012.

**Comment:** P9, L157-158: Is this e-folding time supposed to account for the deposition of smoke along the path? How is this done - is a number on each particle decreased, or do some of the particles actually disappear during the simulation?

**Response:** The e-folding time of 5 days is the mean lifetime of ozone over the Intermountain West accounting for the loss due to dry deposition and chemistry, and it is applied to the mass of the particles in the FLEXPART model.

We state following the previous comment in this section: "Each particle carries a small

account of mass decaying with an e-folding time of 5 days (mean lifetime of ozone in the Intermountain West due to chemical loss and dry deposition as shown in Fiore et al. (2002))."

**Comment:** P10, L169: My understanding is that MISR observations suggest that plumes go above the boundary layer 20-25% of the time, so "often" seems a little vague and misleading.

**Response:** Thanks for the correction. We now state here: "We use 5 km in the vertical because previous studies have shown that fire emissions are occasionally lifted to above the planetary boundary layer and up to 5 km above the surface (Val Martin et al., 2010; Sofiev et al., 2013)".

**Comment:** P10, L170: It is true that 5 km and 5 days generally gave the best correlation, but the change in the fit wasn't very significant compared to 2 km (PBL height) and 5 days.

**Response:** We now state: "as shown in Table S1, it provides slightly better correlations with the OC aerosol concentrations than values with 2 km and 2-4 days."

**Comment:** P10, L175: I'd like to see an equation for variable tr(i,j) as well, that shows how the residence time for a single layer is calculated and how the layers are integrated vertically.

**Response:** We suggest readers refer to previous work of Stohl et al. (2003) and Seibert and Frank (2004) for the detailed calculation of residence time, which is difficult to express in 1-2 equations here.

We state in the text, "tr(i,j)) is FLEXPART calculated daily residence time as described in detail by Stohl et al. (2003) and Seibert and Frank (2004)".

Reference:

Stohl, A., et al.: A backward modeling study of intercontinental pollution transport using aircraft measurements, J. Geophys. Res., 108, doi: 10.1029/2002JD002862 , 2003.

Seibert, P., and Frank, A.: Source-receptor matrix calculation with a Lagrangian particle dispersion model in backward mode, Atmos. Chem. Phys., 4, 51–63, 2004.

**Comment:** P11, L189: Since Table S1 defines the variables, I think it should be moved to the main paper.

**Response:** We now move this Table to the main text (Table 1).

**Comment:** P11, L198: I understand the choice of MLR limits what you can do to look at nonlinearity, but why did you choose the square root of the index instead of, say, the square of the index?

**Response:** We now state: "We also include the square root of FIs and FII (SqrFIs and SqrFII) as variables in the regression model to at least partly account for the non-linearity of ozone chemistry in wildfire plumes, and to narrow the distribution of FI values that are highly episodic."

**Comment:** P11, L202: You should briefly discuss how the model doesn't include interaction terms between the predictors, and the effect this might have on the model performance.

**Response:** Collinearity between predictors can affect the model performance without the interactions terms.

We state in this section: "We do not include the interaction terms to simplify the MLR models. We acknowledge that including FI and meteorological parameters while neglecting their interaction terms in the MLR models inevitably leads to some degree of collinearity. A measure of it is called tolerance (calculated as percent of variance in the predictor that cannot be accounted for by the other predictors) or variance inflation

factors (VIF, the inverse of tolerance), with VIF values greater than 10 suggesting a strong collinearity (Field et al., 2009). Our MLR models for all sites (Section 3) show tolerable VIF values (<5), supporting our approach described above to limit the collinearity."

**Comment:** P14, L236: Do you mean this is just a reanalysis of the Zhang et al. (2014) output, or did you rerun the simulations? You note later that the NOx emissions in this simulation are too high – why didn't you use the lower value here?

**Response:** We have run the model simulation with the standard GEOS-Chem v8-02-03 as presented in Zhang et al. (2014) and a sensitivity simulation with the reduced NOx emission factor. Both simulations are analyzed in this study, and they do not affect our conclusion.

We now state in the text: "We conduct the GEOS-Chem ozone simulations over North America for three-year (2006-2008) using the wildfire area burned of Yue et al. (2013). Zhang et al. (2014) has suggested that wildfire NOx emission factor in the standard GEOS-Chem simulation can be too high by a factor of 3. We thus also conduct a sensitivity simulation with a reduced wildfire NOx emission factor (from 3.0 g to 1.0 g NO per kg of dry mass burned following Zhang et al. (2014))."

**Comment:** P14, L250: I think you mean "all except for GRC" show weaker correlations, or there is an error in Table S2.

**Response:** Values in Table S2 (now Table S1) are correct. We now state: "We also test the correlations of OC aerosols with Fire Index calculated using trajectory residence time at lower altitudes or shorter backward time periods, and they in general show slightly weaker correlations (Table S1)."

**Comment:** P15, L267: Can you explain why you get poorer correlations for Salt Lake City than in the Jaffe et al. (2013) study? What does this imply for your other

results?

**Response:** This is largely because the two studies focused on different months and periods (June-August 1989-2010 in our study vs. June-September 2000-2012 in Jaffe et al).

We now state in the text: "Here we also applied our MLR models to MDA8 ozone concentrations at SLC in the summers 1990-2010. We find FI and meteorological variables can explain 48% of the daily MDA8 ozone variation for summers 1990-2010 (46% if meteorological variables alone are used, and 57% if September data are also considered that explains the higher correlation reported in Jaffe et al. (2013))".

**Comment:** P15, L270: I think this dependence of the performance on altitude makes sense, but a scatter plot of R2 versus site altitude in the supplement would help to prove it.

**Response:** We now add such a figure (scatter-plot of R2 vs. site altitude) in the supplement as Figure S3, and state in the text: "In addition, as shown in Table 2 and Figure S3 the MLR model R2 values for higher-altitude CASTNet sites (> 2000m such as CNT, MEV, PND) are generally lower than values for lower-altitude sites (such as GLR, CHA and BBE)."

**Comment:** P15-16, L287-L290: Since you include the fire index as a predictor, the fact that the residuals don't correlate with TFI just shows that the MLR procedure is working as expected, right? The second clause of this sentence, that underestimates occur even in the absence of fires, seems like more convincing evidence to me.

**Response:** The MLR models are applied to individual daily FI values (rather than TFI) for the whole dataset, and thus the model performances for a particular subset (MDA8 > 70 ppbv) worth emphasizing.

We now state here: "We show in Figure 3 the relationships of TFI values with

measured MDA8 ozone, MLR wildfire ozone enhancements, and MLR residuals to assess the model performance for the subset of high ozone days (MDA8 > 70 ppbv). The MLR model residuals for those high ozone days have little correlation with TFI, and most of the model underestimates occur when there are small fire impacts or fires not captured by the FLEXPART retroplumes."

**Comment:** P21, L367-L371: I didn't understand what you were trying to say here – please elaborate or rephrase?

**Response:** We elaborate in the text: "As we can see here, the wildfire-driven interannual variability (0.3-1.5 ppbv) is much weaker than what can be explained by meteorological parameters (49.4-53.5 ppbv for the regional averaged MLR no wildfire ozone). We suggest that some of the strong correlation between summer mean MLR total ozone concentrations and wildfire activities reflects their common relationships with meteorological parameters such as RH and temperature at the interannual scale, e.g., hot and dry summers would have higher ozone concentrations due to stronger photochemistry as well as more wildfire emissions than cold and wet summers (Figure S4)."

**Comment:** P21, L377: Can you please explain why you chose these percentile ranges?

**Response:** We now state in the text: "The three percentile ranges are used to quantify trends in the low, median, and high windows of summer MDA8 ozone concentration. They also allow us to properly calculate the corresponding mean wildfire ozone contributions to total ozone by using percentile ranges rather than a single percentile. We find similar results when using other percentile ranges (49-51th or 47-53th)."

**Comment:** P24, L420-422: I'd suggest cutting this sentence, as the context of the study is already established in the introduction and this statement is incomplete –

Eulerian model errors are not just about resolution, but about errors in amount, location, and timing of biomass burned, error in emission speciation, errors in chemistry, numerical diffusion errors, etc.

**Response:** We now remove the sentence as suggested.

**Comment:** P25, L435: Make clear how this average R2 is calculated.

**Response:** We now state here: "We show that the MLR models explain 60% (estimated for the ensemble of 13 CASTNet sites) of the variability of MDA8 ozone over the US Intermountain West (16%-59% at individual sites), which is comparable with results from current Eulerian CTMs (R2 = 0.25-0.48 as reported in recent studies)."

**Comment:** Typos and Wording Suggestions:

P7, L112-L113: I'd suggest making this a single list:"ozone, organic carbon (OC) aerosols, meteorological parameters, and wildfire area burned data"

**Response:** Changed as suggested.

**Comment:** P7, L115: Expand CASTNet acronym and provide a little more descriptions than just the website.

**Response:** "CASTNet" is fully expanded when first mentioned in Section 1. We add here: "are accessed from CASTNet, a long-term monitoring network established to assess the trends in air pollution and acid deposition due to emission regulations (http://www.epa.gov/castnet)".

**Comment:** P11, L189: Period should go after the parentheses, not before. **Response:** Changed as suggested.

**Comment:** P14, L244: Figure S2, not S3. **Response:** Changed as suggested.

**Comment:** P14, L247: Need a comma after "strong" **Response:** Changed as suggested.

**Comment:** P16, L280: "as would be expected" delete "it" **Response:** Changed as suggested.

**Comment:** P21, L364: Say "summer mean" to be as clear as possible. **Response:** Changed as suggested.

**Comment:** P22, L389: Don't need comma after "Strode et al." **Response:** Changed as suggested.

**Comment:** P23, L411-412: I suggest putting parentheses around the phrase "accounting for 22% of the summer days" **Response:** Changed as suggested.

**Comment:** P25, L437-438: I suggest cutting everything after the R2 value - these references are already discussed in the main text and do not need to be repeated here. **Response:** Changed as suggested.

**Comment:** P25, L442-443: I don't see much consistency at all between the MLR and GEOS-Chem predictions, so you need to make clearer what consistencies you see.

**Response:** We now state here: "We compare wildfire ozone enhancements estimated by the MLR models with those from the GEOS-Chem CTM for summer 2007. While some consistency is found as reflected by their moderate correlations (r=0.34-0.48, statistically significant p < 0.05), the two methods show rather different patterns."

**Comment:** P27, L474: "model in" instead of "model to" **Response:** Changed as suggested.

**Comment:** P33, L677: Add unit '(m)' of terrain elevations to caption, as it is not on the figure color bar. **Response:** Changed as suggested.

**Comment:** P34, L689: "those from the GEOS-Chem" instead of "those by the GEOS-Chem" **Response:** Changed as suggested.

[Figure]

**Comment:** P38, Figure 6: The wildfire trend values are very hard to see – maybe plot on a secondary y axis? In addition, since the trends are generally not statistically significant perhaps this could be moved to the supplement?

**Response:** We now re-plot the figure with a secondary y-axis representing the wildfire contribution values. We agree that the trends are generally not significant, and have removed relevant statements from the Abstract and Conclusion, also based on another comment above. We think this analysis is valuable to keep in the discussion.

**Comment:** P38, L734: remove "S" from "SMLR" for consistency with the rest of the paper. **Response:** Changed as suggested.

**Comment:** P40, L741: Need a space between "relative humidity" and "(RH)" **Response:** Changed as suggested.

**Comment:** Figure S1, Caption, L3 and 7: "residence time" not "resident time" **Response:** Changed as suggested.

**Comment:** Table S1, Footnotes, L37: Should say "m (PBLH, HGT)", delete the rest. **Response:** Changed as suggested.

**Comment:** Table S1, Footnotes, L38: change to "mean represents the average" **Response:** Changed as suggested.

**Comment:** Table S3, Footnote c: Put the explanation for the bold text in the figure caption, not the footnote. **Response:** Changed as suggested.
* * *

---

## Author Response (AR1)

Dear Editor Zhang:

Please find below our itemized responses to the reviewer's comments and a mark-up manuscript. We have addressed all the comments raised by both reviewers, and incorporated them in the revised manuscript.

Thank you very much for your consideration.

Sincerely,
Xiao Lu, Lin Zhang, et al.
* * *
**Reviewer 1**
Overview: The paper presents a new approach to examining the influence of wildfire smoke on ozone mixing ratios at remote/rural monitoring sites in the U.S. intermountain west. Overall the paper is well written and suitable for publication in ACP. I recommend that the authors consider the following ideas in revising the manuscript.

**Response:**
**We thank the reviewer for the valuable comments. All of them have been implemented in the revised manuscript. Please see our itemized responses below.**

1) Line 285: The sentence staring with "These underestimates" requires substantially more justification/analysis/references.
**Response: We move the original Figure S4 that shows these model underestimates to the main text (Figure 3), and now state:**
**"These underestimates, however, are not likely due to model underestimates of wildfire ozone influences. We show in Figure 3 the relationships of TFI values with measured MDA8 ozone, MLR wildfire ozone enhancements, and MLR residuals to assess the model performance for the subset of high ozone days (MDA8 > 70 ppbv). The MLR model residuals for those high ozone days have little correlation with TFI, and most of the model underestimates occur when there are small fire impacts or fires not captured by the FLEXPART retroplumes. We suggest that these underestimates may be associated with other factors not included in the statistical model such as transport from Asia or California, from lightning emissions or stratosphere. These processes could episodically produce more than 10 ppbv ozone in summer over the US Intermountain West (Zhang et al., 2014)."**

2) Line 315: There are many reasons that a model like GEOS-Chem will not adequately represent the role of fires. The standard versions of GEOS-Chem do not emit short lived VOCs, and the emission factors for NOx emissions from fires are quite variable in reality. The model also adds all the emissions within the boundary layer. The authors clearly recognize this because they use a 5km cut off for the FLEXPART analysis, and are certainly aware of recent work by Val Martin et al. (e.g. 2010) with respect to plume heights over North America. This should be discussed in depth or omitted. A reference to Zhang et al., (2014) is inadequate.

**Response: We agree and add more text discussing why GEOS-Chem may not adequately represent wildfire chemistry.**

**We now state: "We can see that GEOS-Chem simulates up to 40 ppbv wildfire ozone enhancements for the short-distance sites, much higher than the MLR estimates (mean value of 3.96 ppbv versus 1.85 ppbv). A sensitivity simulation with a reduced wildfire NOx emission factor (from 3.0 g to 1.0 g NO per kg of dry mass burned) would decrease the GEOS-Chem mean ozone enhancement for the short-distance sites from 3.96 ppbv to 2.06 ppbv. On the other hand, for the long-distance sites, the GEOS-Chem wildfire ozone enhancements become substantially lower than MLR (0.77 ppbv versus 1.02 ppbv). We see GEOS-Chem largely overestimates wildfire ozone influences near the source regions but fails to capture continued ozone production in wildfire plumes downwind, as also pointed out by Zhang et al. (2014). It reflects the difficulties for Eulerian models such as GEOS-Chem to simulate wildfire ozone production due to, e.g., missing short-lived VOCs (Jaffe and Wigder, 2012), inadequate PAN chemistry (Alvarado et al., 2010; Fischer et al., 2014), and limiting all fire emissions in the boundary layer without considering their injection heights up to the troposphere (Val Martin et al., 2010; Sofiev et al., 2013)."**

3) Why does this paper narrowly focus on the intermountain west? This region has many wildfires, but the smoke travels and the impact on ozone may be larger downwind (see Brey and Fischer, 2016). S. Brey and E.V. Fischer (2016), Smoke in the City: How often and where does smoke impact summertime ozone in the United States, Environ. Sci. Tech., DOI:10.1021/acs.est.5b05218.

**Response: This study follows our previous work of Zhang et al. (2014), which focused on the Intermountain West where background ozone concentrations are high and the ozone trends are not fully understood as we described in the Introduction. It also demonstrates feasibility of our statistical approach to quantify wildfire ozone influences. We expect future work to apply the approach to other regions in the US or over the world.**

**We state in the Conclusion: "A recent study by Brey and Fischer (2016) investigated fire impacts on ozone at urban sites over the contiguous US, and found that fire ozone influences can be even higher at locations with high NOx emissions."**

4) I have two questions with respect to Figure 7 (and the associated discussion). First, is it appropriate to use the entire range of 1989-2010 to look at the number of exceedance days. There have been trends in ozone during this time. Second, and more importantly, would it be more appropriate to view the exceedance days as a percentage of the total, rather than as a count. Yes, there will be more exceedance days as we lower (tighten) the ozone standard, all things held the same. However, do we have a way to determine if the relative importance of fires will increase?

**Response: To answer the two questions: (1) Figure 7 (now Figure 9) has shown the time series of summer ozone exceedance days during 1989-2010. There is no significant trend in the exceedance days with or without wildfire impacts. And (2) we also calculate the percentage of wildfire contributed vs. total exceedance days, but find no increase in the**

**relative importance of fires as lowering the ozone standard.**

**We now state in this section "We find no statistically significant trends in the number of exceedances for both the measured ozone concentrations and ozone in the absence of wildfires during the summers 1989-2010", and "In total, wildfires contribute 28%, 31% and 32% of the days with MDA8 ozone exceeds 65, 70, and 75 ppbv, respectively, reflecting small changes in the relative importance of wildfire influences as lowering the air quality standard over this region."**

5) Finally, I think all the SI materials should be moved into the main paper. There are very important figures in the SI materials, and I had to refer to them to follow the paper. Without them in the main manuscript, it would be easy to overlook the fact that the MLR really does not do a good job reproducing the highest ozone days. This is an important point in considering the value of this analysis.

**Response: We now move the original Figure S3, Figure S4 (now Figure 2 and 3), and Table S1 (now Table 1) to the main text. Figure S3 and S4 explains that the MLR model underestimates of high ozone values are not associated with fire impacts. Tables S1, as also suggested by the Reviewer 2, shall be included in the main text. We think the rest can be kept as SI materials for limiting the length of the paper.**

**Reviewer 2:**
This paper uses back trajectories from the Lagrangian particle dispersion model FLEXPART and estimated fire emissions for the years 1989-2010 to define a Fire Index for 13 CASTNet sites in the Intermountain West. This fire index and various meteorological parameters are used as predictors in a multi-linear regression (MLR) model that predicts daily MDA8 O3 at these sites. The estimated impact of the fire index terms in the model is then used to determine the influence of wildfires on the MDA8 O3, and this estimate is compared to estimated of the Eulerian chemical transport model GEOS-Chem. The authors find that wildfires enhance the summer mean MDA8 O3 by 0.3-1.5 ppbv, with episodic daily increases of 10-20 ppbv at individual sites. They find that GEOS-Chem tends to over-predict the near source formation of O3 and under-predict the downwind formation, consistent with previous Eulerian model studies. Finally, they find that the influence of wildfires is especially important on high O3 days, where 31% of the days with MDA8 O3 over 70 ppbv would not have occurred in the absence of wildfires.

This is a well-done, innovative study and a well-written manuscript. The development of the fire index and the MLR both help to understand the complex influence of fire emissions on O3 in the intermountain west and to expose errors in Eulerian models of this process. The methodology is generally sound and the results are consistent with our understanding of fire chemistry. While I have a few concerns that I would like to see addressed before publication, as detailed below, in general this is a very nice study that should be published.

**Response:**
**We thank the reviewer for the valuable comments. All of them have been implemented in the revised manuscript. Please see our itemized responses below.**

Major Concerns:

I have concerns with two of the conclusions of the paper:

1. The abstract states (P2, L32-33) that wildfires contribute 15% of the measured increasing but statistically insignificant trend in MDA8 O3, and this is also stated in the conclusions section (P26, L461-462). However, as neither trend is statistically significant, I disagree with including the 15% value as a major conclusion of the paper, where it might be erroneously quoted without proper context. Thus I recommend that the abstract and conclusion statements be removed from the paper, but the discussion in Section 4.3 remain, as the trend results are given proper context there.

**Response: We agree that considering the complexity of wildfire impacts and uncertainties in the trends, 15% may not be well constrained. We now remove the statements from the abstract and conclusion, while keeping them in the discussion section.**

2. P20, L357-360 states that the interannual variability in MDA8 O3 appears to be more controlled by interannual variations of the meteorological parameters, as the meteorological variables can account for "most" of the interannual variability in the MDA8 O3, even without fires. I do not think this conclusion is adequately supported by the presented evidence. The fact that the MLR for the met parameters has roughly the same interannual variability as the measurements could be just a statistical artifact of the MLR procedure, with the interannual variability incorrectly accounted for by the meteorological predictors. The conclusion would be more convincing if evidence were presented of the interannual variability of specific meteorological parameters (Temperature, RH, etc.), and if the highs and lows in the summer means of these raw variables were consistent with the highs and lows in MDA8 O3.

**Response: Thanks for pointing it out. To support this conclusion, we now present such a figure in supplement (Figure S4) comparing the interannual variability of relative humidity and temperature with MDA8 ozone.**

**.**

**We also state in the text: "This is further supported by the strong interannual correlations between summer mean MDA8 ozone and meteorological parameters such as daytime mean RH and surface temperature at individual sites and for the regional averages ($r = -0.69$ for RH, $r = 0.48$ for temperature), as shown in Figure S4."**

Minor Concerns:

P7, L122: Please add the latitude-longitude or ID number for the SalLake City site you are using.

**Response: We now state here "we use hourly ozone measurements from 1990-2010 at the Salt Lake City (SLC, 40.6N, 111.9W, 1300m) urban site (data available at https://www3.epa.gov/airdata/) for comparison with the CASTNet background sites and**

the previous work of Jaffe et al. (2013)."

P9, L156: 250,000 is a huge number of particles to track, and is probably overkill. Usually 500 particles per receptor (time step and location) is sufficient. How many time steps are there each day, and how many particles are released in each time step?
**Response: The particle number is selected to ensure that model calculated retroplumes are statistically robust. It is in the middle of two previous studies using the FLEXPART model: 40000 in Cooper et al. (2010) and 1 million in Stohl et al. (2012).**

**To address this comment, we now state in the text: "For each day at a receptor site, FLEXPART was run in backward mode, with 250,000 particles released at the site location at a constant hourly rate (~10k particles per hour) during the first 24 hours. Previous studies have used the particle sizes of 40000 (Cooper et al., 2010) and 1 million (Stohl et al., 2012) represent a retroplume."**

P11, L189: Since Table S1 defines the variables, I think it should be moved to the main paper.
**Response: We now move this Table to the main text (Table 1).**

P11, L198: I understand the choice of MLR limits what you can do to look at nonlinearity, but why did you choose the square root of the index instead of, say, the square of the index?
**Response: We now state: "We also include the square root of $FI_s$ and $FI_l$ ($SqrFI_s$ and $SqrFI_l$) as variables in the regression model to at least partly account for the non-linearity of ozone chemistry in wildfire plumes, and to narrow the distribution of FI values that are highly episodic."**

P11, L202: You should briefly discuss how the model doesn't include interaction terms between the predictors, and the effect this might have on the model performance.
**Response: Collinearity between predictors can affect the model performance without the interactions terms.**
**We state in this section "We do not include the interaction terms to simplify the MLR models. We acknowledge that including FI and meteorological parameters while neglecting their interaction terms in the MLR models inevitably leads to some degree of collinearity. A measure of it is called tolerance (calculated as percent of variance in the predictor that cannot be accounted for by the other predictors) or variance inflation factors (VIF, the inverse of tolerance), with VIF values greater than 10 suggesting a strong collinearity (Field et al., 2009). Our MLR models for all sites (Section 3) show tolerable VIF values (<5), supporting our approach described above to limit the collinearity."**

P14, L236: Do you mean this is just a reanalysis of the Zhang et al. (2014) output, or did you rerun the simulations? You note later that the NOx emissions in this simulation are too high – why didn't you use the lower value here?

**Response: We have run the model simulation with the standard GEOS-Chem v8-02-03 as presented in Zhang et al. (2014) and a sensitivity simulation with the reduced NOx emission factor. Both simulations are analyzed in this study, and they do not affect our conclusion.**

**We now state in the text: "We conduct the GEOS-Chem ozone simulations over North America for three-year (2006-2008) using the wildfire area burned of Yue et al. (2013). Zhang et al. (2014) has suggested that wildfire NOx emission factor in the standard GEOS-Chem simulation can be too high by a factor of 3. We thus also conduct a sensitivity simulation with a reduced wildfire NOx emission factor (from 3.0 g to 1.0 g NO per kg of dry mass burned following Zhang et al. (2014))."**

P14, L250: I think you mean "all except for GRC" show weaker correlations, or there is an error in Table S2.
**Response: Values in Table S2 (now Table S1) are correct. We now state: "We also test the correlations of OC aerosols with Fire Index calculated using trajectory residence time at lower altitudes or shorter backward time periods, and they in general show slightly weaker correlations (Table S1)."**

P15, L267: Can you explain why you get poorer correlations for Salt Lake City than in the Jaffe et al. (2013) study? What does this imply for your other results?
**Response: This is largely because the two studies focused on different months and periods (June-August 1989-2010 in our study vs. June-September 2000-2012 in Jaffe et al).**
**We now state in the text: "Here we also applied our MLR models to MDA8 ozone concentrations at SLC in the summers 1990-2010. We find FI and meteorological variables can explain 48% of the daily MDA8 ozone variation for summers 1990-2010 (46% if meteorological variables alone are used, and 57% if September data are also considered that explains the higher correlation reported in Jaffe et al. (2013))".**

P15, L270: I think this dependence of the performance on altitude makes sense, but a scatter plot of R2 versus site altitude in the supplement would help to prove it.
**Response: We now add such a figure (scatter-plot of R2 vs. site altitude) in the supplement as Figure S3, and state in the text:**
**"In addition, as shown in Table 2 and Figure S3 the MLR model $R^2$ values for higher-altitude CASTNet sites (> 2000m such as CNT, MEV, PND) are generally lower than values for lower-altitude sites (such as GLR, CHA and BBE)."**

P15-16, L287-L290: Since you include the fire index as a predictor, the fact that the residuals don't correlate with TFI just shows that the MLR procedure is working as expected, right? The second clause of this sentence, that underestimates occur even in the absence of fires, seems like more convincing evidence to me.
**Response: The MLR models are applied to individual daily FI values (rather than TFI) for the whole dataset, and thus the model performances for a particular subset (MDA8 >**

**70 ppbv) worth emphasizing.**

**We now state here: "We show in Figure 3 the relationships of TFI values with measured MDA8 ozone, MLR wildfire ozone enhancements, and MLR residuals to assess the model performance for the subset of high ozone days (MDA8 > 70 ppbv). The MLR model residuals for those high ozone days have little correlation with TFI, and most of the model underestimates occur when there are small fire impacts or fires not captured by the FLEXPART retroplumes."**

P21, L367-L371: I didn't understand what you were trying to say here – please elaborate or rephrase?

**Response: We elaborate in the text: "As we can see here, the wildfire-driven interannual variability (0.3-1.5 ppbv) is much weaker than what can be explained by meteorological parameters (49.4-53.5 ppbv for the regional averaged MLR no wildfire ozone). We suggest that some of the strong correlation between summer mean MLR total ozone concentrations and wildfire activities reflects their common relationships with meteorological parameters such as RH and temperature at the interannual scale, e.g., hot and dry summers would have higher ozone concentrations due to stronger photochemistry as well as more wildfire emissions than cold and wet summers (Figure S4)."**

P21, L377: Can you please explain why you chose these percentile ranges?

**Response: We now state in the text: "The three percentile ranges are used to quantify trends in the low, median, and high windows of summer MDA8 ozone concentration. They also allow us to properly calculate the corresponding mean wildfire ozone contributions to total ozone by using percentile ranges rather than a single percentile. We find similar results when using other percentile ranges (49-51th or 47-53th)."**

P24, L420-422: I'd suggest cutting this sentence, as the context of the study is already established in the introduction and this statement is incomplete – Eulerian model errors are not just about resolution, but about errors in amount, location, and timing of biomass burned, error in emission speciation, errors in chemistry, numerical diffusion errors, etc.

**Response: We now remove the sentence as suggested.**

P25, L435: Make clear how this average R2 is calculated.

**Response: We now state here: "We show that the MLR models explain 60% (estimated for the ensemble of 13 CASTNet sites) of the variability of MDA8 ozone over the US Intermountain West (16%-59% at individual sites), which is comparable with results from current Eulerian CTMs ($R^2$ = 0.25-0.48 as reported in recent studies)."**

Typos and Wording Suggestions:

P7, L112-L113: I'd suggest making this a single list:"ozone, organic carbon (OC) aerosols, meteorological parameters, and wildfire area burned data"

**Response: Changed as suggested.**

P7, L115: Expand CASTNet acronym and provide a little more descriptions than just the website.
**Response: "CASTNet" is fully expanded when first mentioned in Section 1. We add here: "are accessed from CASTNet, a long-term monitoring network established to assess the trends in air pollution and acid deposition due to emission regulations (http://www.epa.gov/castnet)".**

P11, L189: Period should go after the parentheses, not before.
**Response: Changed as suggested.**

P14, L244: Figure S2, not S3.
**Response: Changed as suggested.**

P14, L247: Need a comma after "strong"
**Response: Changed as suggested.**

P16, L280: "as would be expected" delete "it"
**Response: Changed as suggested.**

P21, L364: Say "summer mean" to be as clear as possible.
**Response: Changed as suggested.**

P22, L389: Don't need comma after "Strode et al."
**Response: Changed as suggested.**

P23, L411-412: I suggest putting parentheses around the phrase "accounting for 22% of the summer days"
**Response: Changed as suggested.**

P25, L437-438: I suggest cutting everything after the R2 value - these references are already discussed in the main text and do not need to be repeated here.
**Response: Changed as suggested.**

P25, L442-443: I don't see much consistency at all between the MLR and GEOS-Chem predictions, so you need to make clearer what consistencies you see.
**Response: We now state here: "We compare wildfire ozone enhancements estimated by the MLR models with those from the GEOS-Chem CTM for summer 2007. While some consistency is found as reflected by their moderate correlations (r=0.34-0.48, statistically significant p < 0.05), the two methods show rather different patterns."**

P27, L474: "model in" instead of "model to"
**Response: Changed as suggested.**

P33, L677: Add unit '(m)' of terrain elevations to caption, as it is not on the figure color bar.

**Response: Changed as suggested.**

P34, L689: "those from the GEOS-Chem" instead of "those by the GEOS-Chem"
**Response: Changed as suggested.**

P38, Figure 6: The wildfire trend values are very hard to see – maybe plot on a secondary y axis? In addition, since the trends are generally not statistically significant perhaps this could be moved to the supplement?
**Response: We now re-plot the figure with a secondary y-axis representing the wildfire contribution values. We agree that the trends are generally not significant, and have removed relevant statements from the Abstract and Conclusion, also based on another comment above. We think this analysis is valuable to keep in the discussion.**

P38, L734: remove "S" from "SMLR" for consistency with the rest of the paper.
**Response: Changed as suggested.**

P40, L741: Need a space between "relative humidity" and "(RH)"
**Response: Changed as suggested.**

Figure S1, Caption, L3 and 7: "residence time" not "resident time"
**Response: Changed as suggested.**

Table S1, Footnotes, L37: Should say "m (PBLH, HGT)", delete the rest.
**Response: Changed as suggested.**

Table S1, Footnotes, L38: change to "mean represents the average"
**Response: Changed as suggested.**

Table S3, Footnote c: Put the explanation for the bold text in the figure caption, not the footnote.
**Response: Changed as suggested.**

[revised manuscript text omitted]

WSPsurf
RH
SRAD | Daytime mean[b] surface temperature
Daytime mean wind speed
Daytime mean relative humidity
Daytime mean solar radiation | CASTNet surface monitoring sites in the U.S. Intermountain West (http://www.epa.gov/castnet), for 13 CASTNet sites only |
| Tmax

AWND | Daily maximum temperature

Daily average daily wind speed | NOAA, National Climatic Data Center: Climate Data Online (http://www.ncdc.noaa.gov/cdo-web/), for Salt Lake City urban site only |
| PBLH | Gridded daily maximum planetary boundary height | NCEP Climate Forecast System Reanalysis (http://rda.ucar.edu/datasets/ds093.0/) |
| PRCP | Gridded daily precipitation | Climate Prediction Center of the National Weather Service (ftp://ftp.cpc.ncep.noaa.gov/precip/CPC_UNI_PRCP/GAUGE_CONUS/V1.0/) |
| U
V
WSP
Ome
SH
HGT
T
dT | Gridded daily mean 850, 700, 500 hPa zonal wind
Gridded daily mean 850, 700, 500 hPa meridional wind
Gridded daily mean 850, 700, 500 hPa horizontal wind
Gridded daily mean 850, 700, 500 hPa vertical velocity
Gridded daily mean 850, 700, 500 hPa specific humidity
Gridded daily mean 850, 700, 500 hPa geopotential heights
Gridded daily mean 850, 700, 500 hPa temperature
Gridded daily mean temperature at 1000mb minus that at 850 hPa | NCEP/NCAR Reanalysis dataset (http://www.esrl.noaa.gov/psd/data/timeseries/daily/) |

[a]Units are °C (Tsurf, T, dT, Tmax), m s$^{-1}$ (WSPsurf, WSP, U, V, AWND), % (RH), W m$^{-2}$ (SRAD), m (PBLH, HGT), kg·kg$^{-1}$ (SH), 0.1 mm (PRCP) , and pa s$^{-1}$ (Ome).

[b]Daytime mean represents the average for 10:00-17:00 local time.

**Table 2**. Multiple linear regression (MLR) models for summer MDA8 ozone at 13 Intermountain West CASTNet sites[a]

| Sites[b] | $R^2$ (N) | Variables included in the MLR model[c] |
| --- | --- | --- |
| **Glacier NP, MT** (GLR, 48N, 113W, 976m) | 0.59 (1809) | RH, WSPsurf, SRAD, U, V, Ome, SH, HGT, T, dT, SH, SqrFI$_l$, SqrFI$_s$, |
| **Yellowstone NP, WY** (YEL, 44N, 110W, 2400m) | 0.35 (1611) | RH, WSPsurf, Tsurf, SRAD, U, V,WSP, OME, HGT, T, dT, SH, SqrFI$_l$, SqrFI$_s$, FI$_l$, |
| **Pinedale, WY** (PND, 42N, 109W, 2388m) | 0.28 (1888) | RH, WSPsurf, Tsurf, SRAD, U, V, WSP, Ome, HGT, T, SH, SqrFI$_l$, SqrFI$_s$, FI$_s$, |
| **Centennial, WY** (CNT, 41N, 106W, 3178m) | 0.19 (1925) | RH, U, WSP, HGT, T, SH, SqrFI$_l$, SqrFI$_s$, FI$_s$, |
| **Rocky Mtn NP, CO** (ROM, 40N, 105W, 2743m) | 0.36 (1367) | RH, WSPsurf, Tsurf, SRAD, PRCP, U, Ome, T, SH, FI$_s$, SqrFI$_l$, SqrFI$_s$, |
| **Gothic, CO** (GTH, 38N, 106W, 2926m) | 0.29 (1906) | RH, WSPsurf, U, V, WSP, Ome, HGT, T, dT, SH, SqrFI$_l$, FI$_l$, |
| **Mesa Verde NP, CO** (MEV, 37N, 108W, 2165m) | 0.23 (1321) | RH, WSPsurf, Tsurf, SRAD, U, V, T, dT, SqrFI$_l$, SqrFI$_s$, |
| **Great Basin NP, NV** (GRB, 39N, 114W, 2060m) | 0.40 (1360) | WSPsurf, Tsurf, SRAD, U, WSP,Ome, SH, Ome, HGT, SH, SqrFI$_l$, SqrFI$_s$, FI$_s$, |
| **Canyonlands NP, UT** (CAN, 38N, 109W, 1809m) | 0.16 (1379) | RH, WSPsurf, Tsurf, V, Ome, T, FI$_l$, SqrFI$_l$, SqrFI$_s$, |
| **Grand Canyon NP, AZ** (GRC, 36N, 112W, 2073m) | 0.34 (1912) | RH, WSPsurf, SRAD, PRCP, U, V, WSP, Ome, HGT, T, SH, SqrFI$_l$, FI$_l$, |
| **Petrified Forest, AZ** (PET, 34N, 109W, 1723m) | 0.43 (654) | RH, SRAD, V, WSP, Ome, HGT, T, dT, SH, SqrFI$_l$ |
| **Chiricahua NM, AZ** (CHA, 32N, 109W, 1570m) | 0.50 (1754) | RH, SRAD, PBLH, U, V, WSP, HGT, T, dT, SH, SqrFI$_l$, FI$_l$, |
| **Big Bend NP, TX** (BBE, 29N, 103W, 1052m) | 0.46 (1196) | RH, WSPsurf, SRAD, U, V, WSP, HGT, T, SqrFI$_l$, SqrFI$_s$, FI$_l$, FI$_s$ |

[a] Coefficients of determination ($R^2$), sample numbers (N), and variables included in the MLR models.

[b] NP = National Park, NM = National Monument, MT = Montana, WY = Wyoming, CO = Colorado, NV= Nevada, UT = Utah, AZ = Arizona, TX = Texas.

[c] Fire Index (FI$_l$, FI$_s$), square root of FI (SqrFI$_l$, SqrFI$_s$), and meteorological parameters including (1) surface measurements: daytime (10:00-17:00 local time) mean temperature (Tsurf), wind speed (WSPsurf), relative humidity (RH), and solar radiation flux (SRAD); (2) gridded daily precipitation (PRCP); (3) NCEP data at 850/700/500 hPa pressure levels: daily maximum planetary boundary layer height (PBLH), daily mean zonal wind speed (U), meridional wind speed (V), horizontal wind speed (WSP), temperature(T), geopotential height (HGT), vertical velocity (Ome), specific humidity (SH), and temperature at 1000hPa minus that at 850 hPa (dT). Please refer to Table S1 and S3 for details on the parameters and MLR models.